# A Ranking Game for Imitation Learning

**Harshit Sikchi**                                    *hsikchi@utexas.edu*
*Department of Computer Science, The University of Texas at Austin*

**Akanksha Saran**                              *akanksha.saran@microsoft.com*
*Microsoft Research NYC*

**Wonjoon Goo**                                   *wonjoon@cs.utexas.edu*
*Department of Computer Science, The University of Texas at Austin*

**Scott Niekum**                                  *sniekum@cs.umass.edu*
*Department of Computer Science, University of Massachusetts Amherst*

**Reviewed on OpenReview:** *https://openreview.net/forum?id=d3rHk4VAf0*

## Abstract

We propose a new framework for imitation learning—treating imitation as a *two-player ranking-based game* between a policy and a reward. In this game, the reward agent learns to satisfy pairwise performance rankings between behaviors, while the policy agent learns to maximize this reward. In imitation learning, near-optimal expert data can be difficult to obtain, and even in the limit of infinite data cannot imply a total ordering over trajectories as preferences can. On the other hand, learning from preferences alone is challenging as a large number of preferences are required to infer a high-dimensional reward function, though preference data is typically much easier to collect than expert demonstrations. The classical inverse reinforcement learning (IRL) formulation learns from expert demonstrations but provides no mechanism to incorporate learning from offline preferences and vice versa. We instantiate the proposed ranking-game framework with a novel ranking loss giving an algorithm that can simultaneously learn from expert demonstrations and preferences, gaining the advantages of both modalities. Our experiments show that the proposed method achieves state-of-the-art sample efficiency and can solve previously unsolvable tasks in the Learning from Observation (LfO) setting. Project video and code can be found at this URL.

## 1 Introduction

Reinforcement learning relies on environmental reward feedback to learn meaningful behaviors. Reward specification is a hard problem (Krakovna, 2018), thus motivating imitation learning (IL) as a technique to bypass reward specification and learn from expert data, often via Inverse Reinforcement Learning (IRL) techniques. Imitation learning typically deals with the setting of Learning from Demonstrations (LfD), where expert states and actions are provided to the learning agent. A more practical problem in imitation learning is Learning from Observations (LfO), where the learning agent has access to only the expert observations. This setting is common when access to expert actions are unavailable such as when learning from accessible observation sources like videos or learning to imitate across different agent morphologies. We note that LfD and LfO settings differ from the setting where the agent has access to the environment reward function along with expert transitions, referred to as Reinforcement Learning from Demonstrations (RLfD) (Jing et al., 2020; Zhang et al., 2020; Brys et al., 2015).

Learning to imitate using expert observations alone can require efficient exploration when the expert actions are unavailable as in LfO (Kidambi et al., 2021). Incorporating preferences over potentially suboptimal trajectories for reward learning can help reduce the exploration burden by regularizing the reward function

and providing effective guidance for policy optimization. Previous literature in learning from preferences either assumes no environment interaction (Brown et al., 2019; 2020a) or assumes an active query framework with a restricted reward class (Palan et al., 2019). The classical IRL formulation suffers from two issues: (1) Learning from expert demonstrations and learning from preferences/rankings provide complementary advantages for increasing learning efficiency (Ibarz et al., 2018; Palan et al., 2019); however, existing IRL methods that learn from expert demonstrations provide no mechanisms to incorporate offline preferences and vice versa. (2) Optimization is difficult, making the learning sample inefficient (Arenz & Neumann, 2020; Ho & Ermon, 2016) due to the adversarial min-max game.

Our primary contribution is an algorithmic framework casting imitation learning as a ranking game which addresses both of the above issues in IRL. This framework treats imitation as a ranking game between two agents: a reward agent and a policy agent—the reward agent learns to satisfy pairwise performance rankings between different *behaviors* represented as state-action or state visitations, while the policy agent maximizes its performance under the learned reward function. The ranking game is detailed in Figure 1 and is specified by three components: (1) The dataset of pairwise behavior rankings, (2) A ranking loss function, and (3) An optimization strategy. This game encompasses a large subset of both inverse reinforcement learning (IRL) methods and methods which learn from suboptimal offline preferences. Popular IRL methods such as GAIL, AIRL, $f$-MAX (Ho & Ermon, 2016; Ghasemipour et al., 2020; Ke et al., 2021) are instantiations of this ranking game in which rankings are given only between the learning agent and the expert, and a gradient descent ascent (GDA) optimization strategy is used with a ranking loss that maximizes the performance gap between the behavior rankings.

The ranking loss used by the prior IRL approaches is specific to the comparison of optimal (expert) vs. suboptimal (agent) data, and precludes incorporation of comparisons among suboptimal behaviors. In this work, we instantiate the ranking game by proposing a new ranking loss ($L_k$) that facilitates incorporation of rankings over suboptimal trajectories for reward learning. Our theoretical analysis reveals that the proposed ranking loss results in a bounded performance gap with the expert that depends on a controllable hyperparameter. Our ranking loss can also ease policy optimization by supporting data augmentation to make the reward landscape smooth and allowing control over the learned reward scale. Finally, viewing our ranking game in the Stackelberg game framework (see Section 3)—an efficient setup for solving general-sum games—we obtain two algorithms with complementary benefits in non-stationary environments depending on which agent is set to be the leader.

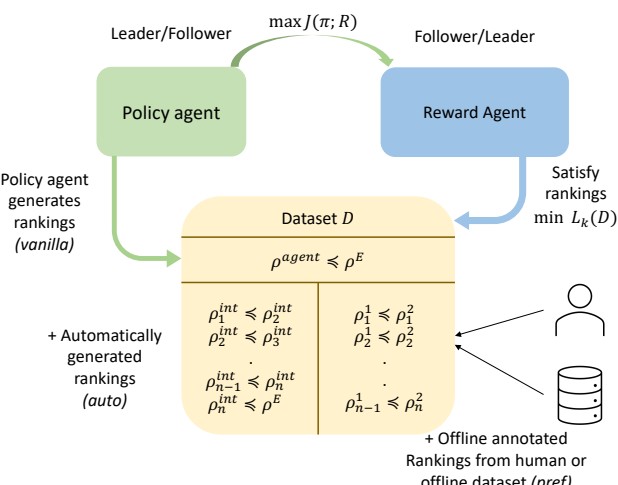

In summary, this paper formulates a new framework `rank-game` for imitation learning that allows us to view learning from preferences and demonstrations under a unified perspective. We instantiate the framework with a principled ranking loss that can naturally incorporate rankings provided by diverse sources. Finally, by incorporating additional rankings—auto-generated or offline—our method: (a) outperforms state-of-the-art methods for imitation learning in several MuJoCo simulated domains by a significant margin and (b) solves complex tasks like imitating to reorient a pen with dextrous manipulation using only a few observation trajectories that none of the previous LfO baselines can solve.

Figure 1: `rank-game`: The Policy agent maximizes the reward function by interacting with the environment. The Reward agent satisfies a set of behavior rankings obtained from various sources: generated by the policy agent (vanilla), automatically generated (auto), or offline annotated rankings obtained from a human or offline dataset (pref). Treating this game in the Stackelberg framework leads to either Policy being a leader and Reward being a follower, or vice-versa.

| IL Method | Offline Preferences | Expert Data | Ranking Loss | Reward Function | Active Human Query |
|---|---|---|---|---|---|
| MaxEntIRL, AdRIL,GAN-GCL, GAIL,$f$-MAX, AIRL | ✗ | LfD | supremum | non-linear | ✗ |
| BCO,GAIfO, DACfO, OPOLO,$f$-IRL | ✗ | LfO | supremum | non-linear | ✗ |
| TREX, DREX | ✓ | ✗ | Bradley-Terry | non-linear | ✗ |
| BREX | ✓ | ✗ | Bradley-Terry | linear | ✗ |
| DemPref | ✓ | LfO/LfD | Bradley-Terry | linear | ✓ |
| Ibarz et al. (2018) | ✓ | LfD | Bradley-Terry | non-linear | ✓ |
| rank-game | ✓ | LfO/LfD | $L_k$ | non-linear | ✗ |

Table 1: A summary of IL methods demonstrating the data modalities they can handle (expert data and/or preferences), the ranking-loss functions they use, the assumptions they make on reward function, and whether they require availability of an external agent to provide preferences during training. We highlight whether a method enables LfD, LfO, or both when it is able to incorporate expert data.

## 2 Related Work

Imitation learning methods are broadly divided into two categories: Behavioral cloning (Pomerleau, 1991; Ross et al., 2011) and Inverse Reinforcement Learning (IRL) (Ng et al., 2000; Abbeel & Ng, 2004; Ziebart et al., 2008; Finn et al., 2016; Fu et al., 2017; Ho & Ermon, 2016; Ghasemipour et al., 2020). Our work focuses on developing a new framework in the setting of IRL through the lens of ranking. Table 1 shows a comparison of the proposed rank-game method to prior works.

**Classical Imitation Game for IRL**: The classical imitation game for IRL aims to solve the adversarial *min-max* problem of finding a policy that minimizes the worst-case performance gap between the agent and the expert. A number of previous works (Ghasemipour et al., 2020; Swamy et al., 2021; Ke et al., 2021) have focused on analyzing the properties of this *min-max* game and its relation to divergence minimization. Under some additional regularization, this *min-max* objective can be understood as minimizing a certain $f$-divergence (Ho & Ermon, 2016; Ghasemipour et al., 2020; Ke et al., 2021) between the agent and expert state-action visitation. More recently, Swamy et al. (2021) showed that all forms of imitation learning (BC and IRL) can be understood as performing moment matching under differing assumptions. In this work, we present a new perspective on imitation in which the reward function is learned using a dataset of behavior comparisons, generalizing previous IRL methods that learn from expert demonstrations and additionally giving the flexibility to incorporate rankings over suboptimal behaviors.

**Learning from Preferences and Suboptimal Data**: Learning from preferences and suboptimal data is important when expert data is limited or hard to obtain. Preferences (Akrour et al., 2011; Wilson et al., 2012; Sadigh et al., 2017; Christiano et al., 2017; Palan et al., 2019; Cui et al., 2021) have the advantage of providing guidance in situations expert might not get into, and in the limit provides full ordering over trajectories which expert data cannot. A previous line of work (Brown et al., 2019; 2020b;a; Chen et al., 2020) has studied this setting and demonstrated that offline rankings over suboptimal behaviors can be effectively leveraged to learn a reward function. Christiano et al. (2017); Palan et al. (2019); Ibarz et al. (2018) studied the question of learning from preferences in the setting when a human is available to provide online preferences[1] (active queries), while Palan et al. (2019) additionally assumed the reward to be linear in known features. Our work makes no such assumptions and allows for integrating offline preferences and expert demonstrations under a common framework.

**Learning from Observation** (LfO): LfO is the problem setting of learning from expert observations. This is typically more challenging than the traditional learning from demonstration setting (LfD), because actions taken by the expert are unavailable. LfO is broadly formulated using two objectives: state-next state marginal matching (Torabi et al., 2019; Zhu et al., 2020b; Sun et al., 2019) and direct state marginal matching (Ni et al., 2020; Liu et al., 2019). Some prior works (Torabi et al., 2018a; Yang et al., 2019; Edwards et al., 2019) approach LfO by inferring expert actions through a learned inverse dynamics model. These methods assume

---

[1] We will use preferences and ranking interchangebly

injective dynamics and suffer from compounding errors when the policy is deployed. A recently proposed method OPOLO (Zhu et al., 2020b) derives an upper bound for the LfO objective which enables it to utilize off-policy data and increase sample efficiency. Our method outperforms baselines including OPOLO, by a significant margin.

## 3 Background

We consider a learning agent in a Markov Decision Process (MDP) (Puterman, 2014; Sutton & Barto, 2018) which can be defined as a tuple: $\mathcal{M} = (\mathcal{S}, \mathcal{A}, P, R, \gamma, \rho_0)$, where $\mathcal{S}$ and $\mathcal{A}$ are the state and action spaces; $P$ is the state transition probability function, with $P(s'|s, a)$ indicating the probability of transitioning from $s$ to $s'$ when taking action $a$; $R : \mathcal{S} \times \mathcal{A} \to \mathbb{R}$ is the reward function bounded in $[0, R_{max}]$; We consider MDPs with infinite horizon, with the discount factor $\gamma \in [0, 1]$, though our results extend to finite horizons as well; $p_0$ is the initial state distribution. We use $\Pi$ and $\mathcal{R}$ to denote the space of policies and reward functions respectively. A reinforcement learning agent aims to find a policy $\pi : \mathcal{S} \to \mathcal{A}$ that maximizes its expected return, $J(R; \pi) = \frac{1}{1-\gamma} \mathbb{E}_{(s,a) \sim \rho^\pi(s,a)}[R(s, a)]$, where $\rho^\pi(s, a)$ is the stationary state-action distribution induced by $\pi$. In imitation learning, we are provided with samples from the state-action visitation of the expert $\rho^{\pi_E}(s, a)$ but the reward function of the expert, denoted by $R_{gt}$, is unknown. We will use $\rho^E(s, a)$ as a shorthand for $\rho^{\pi_E}(s, a)$.

**Classical Imitation Learning**: The goal of imitation learning is to close the imitation gap $J(R_{gt}; \pi^E) - J(R_{gt}; \pi)$ defined with respect to the unknown expert reward function $R_{gt}$. Several prior works (Ho & Ermon, 2016; Swamy et al., 2021; Kostrikov et al., 2019; Ni et al., 2020) tackle this problem by minimizing the imitation gap on all possible reward hypotheses. This leads to a zero-sum (min-max) game formulation of imitation learning in which a policy is optimized with respect to the reward function that induces the largest imitation gap:

$$\texttt{imit-game}(\pi) = \arg\min_{\pi \in \Pi} \sup_{f \in \mathcal{R}} \mathbb{E}_{\rho^E(s,a)}[f(s, a)] - \mathbb{E}_{\rho^\pi(s,a)}[f(s, a)]. \tag{1}$$

Here, assuming realizability ($R_{gt} \in \mathcal{R}$), the imitation gap is upper bounded as follows ($\forall \pi$):

$$J(R_{gt}; \pi^E) - J(R_{gt}; \pi) \leq \sup_{f \in \mathcal{R}} \frac{1}{1-\gamma} [\mathbb{E}_{\rho^E(s,a)}[f(s, a)] - \mathbb{E}_{\rho^\pi(s,a)}[f(s, a)]]. \tag{2}$$

Note that, when the performance gap is maximized between the expert $\pi^E$ and the agent $\pi$, we can observe that the worst-case reward function $f_\pi$ induces a ranking between policy behaviors based on their performance: $\rho^E \succeq \rho^\pi := \mathbb{E}_{\rho^E(s,a)}[f_\pi(s, a)] \geq \mathbb{E}_{\rho^\pi(s,a)}[f_\pi(s, a)], \forall \pi$. Therefore, we can regard the above loss function that maximizes the performance gap (Eq. 2) as an instantiation of the ranking-loss. We will refer to the implicit ranking between agent and the expert $\rho^E \succeq \rho^\pi$ as vanilla rankings and this variant of the ranking-loss function as the *supremum-loss*.

**Stackelberg Games**: A Stackelberg game (Başar & Olsder, 1998) is a general-sum game between two agents where one agent is set to be the leader and the other a follower. The leader in this game optimizes its objective under the assumption that the follower will choose the best response for its own optimization objective. More concretely, assume there are two players $A$ and $B$ with parameters $\theta_A, \theta_B$ and corresponding losses $\mathcal{L}_A(\theta_A, \theta_B)$ and $\mathcal{L}_B(\theta_A, \theta_B)$. A Stackelberg game solves the following bi-level optimization when $A$ is the leader and $B$ is the follower: $\min_{\theta_A} \mathcal{L}_A(\theta_A, \theta_B^*(\theta_A))$ s.t $\theta_B^*(\theta_A) = \arg\min_\theta \mathcal{L}_B(\theta_A, \theta)$. Rajeswaran et al. (2020) showed that casting model-based RL as an approximate Stackelberg game leads to performance benefits and reduces training instability in comparison to the commonly used GDA (Schäfer & Anandkumar, 2019) and Best Reponse (BR) (Cesa-Bianchi & Lugosi, 2006) methods. Fiez et al. (2019); Zheng et al. (2021) prove convergence of Stackelberg games under smooth player cost functions and show that they reduce the cycling behavior to find an equilibrium and allow for better convergence.

## 4 A Ranking Game for Imitation Learning

In this section, we first formalize the notion of the proposed two-player general-sum ranking game for imitation learning. We then propose a practical instantiation of the ranking game through a novel ranking-loss ($L_k$).

---

**Algorithm 1** Meta algorithm: `rank-game` (vanilla) for imitation

---

1:  Initialize policy $\pi_\theta^0$, reward funtion $R_\phi$, empty dataset $\mathcal{D}^\pi$. empirical expert data $\hat{\rho}^E$
2:  **for** $t = 1..T$ iterations **do**
3:    Collect empirical visitation data $\hat{\rho}^{\pi_\theta^t}$ with $\pi_\theta^t$ in the environment. Set $\mathcal{D}^\pi = \{(\hat{\rho}^\pi \preceq \hat{\rho}^E)\}$
4:    Train reward $R_\phi$ to satisfy rankings in $\mathcal{D}^\pi$ using ranking loss $L_k$ in equation 3.
5:    Optimize policy under the reward function: $\pi_\theta^{t+1} \leftarrow \text{argmax}_{\pi'} J(R_\phi; \pi')$
6:  **end for**

---

The proposed ranking game gives us the flexibility to incorporate additional rankings—both auto-generated (a form of data augmentation mentioned as 'auto' in Fig. 1) and offline ('pref' in Fig. 1)—which improves learning efficiency. Finally, we discuss the Stackelberg formulation for the two-player ranking game and discuss two algorithms that naturally arise depending on which player is designated as the leader.

### 4.1   The Two-Player Ranking Game Formulation

We present a new framework, `rank-game`, for imitation learning which casts it as a general-sum *ranking game* between two players — a reward and a policy.

$$\underbrace{\text{argmax}_{\pi\in\Pi} J(R;\pi)}_{\text{Policy Agent}} \quad \underbrace{\text{argmin}_{R\in\mathcal{R}} L(\mathcal{D}^p; R)}_{\text{Reward Agent}}$$

In this formulation, the policy agent maximizes the reward by interacting with the environment, and the reward agent attempts to find a reward function that satisfies a set of pairwise behavior rankings in the given dataset $\mathcal{D}^p$; a reward function satisfies these rankings if $\mathbb{E}_{\rho^{\pi^i}}[R(s,a)] \leq \mathbb{E}_{\rho^{\pi^j}}[R(s,a)], \ \forall \rho^{\pi^i} \preceq \rho^{\pi^j} \in \mathcal{D}^p$, where $\rho^{\pi^i}, \rho^{\pi^j}$ can be state-action or state vistitations.

The dataset of pairwise behavior rankings $\mathcal{D}^p$ can be comprised of the implicit 'vanilla' rankings between the learning agent and the expert's policy behaviors ($\rho^\pi \preceq \rho^E$), giving us the classical IRL methods when a specific ranking loss function – *supremum-loss* is used  (Ho & Ermon, 2016; Ghasemipour et al., 2020; Ke et al., 2021). If rankings are provided between trajectories, they can be reduced to the equivalent ranking between the corresponding state-action/state visitations. In the case when $\mathcal{D}^p$ comprises purely of offline trajectory performance rankings then, under a specific ranking loss function (*Luce-shepard*), the ranking game reduces to prior reward inference methods like T-REX (Brown et al., 2019; 2020b;a; Chen et al., 2020). Thus, the ranking game affords us a broader perspective of imitation learning, going beyond only using expert demonstrations.

### 4.2   Ranking Loss $L_k$ for the Reward Agent

We use a *ranking-loss* to train the reward function—an objective that minimizes the distortion (Iyer & Bilmes, 2012) between the ground truth ranking for a pair of entities $\{x, y\}$ and rankings induced by a parameterized function $R : \mathcal{X} \rightarrow \mathbb{R}$ for a pair of scalars $\{R(x), R(y)\}$. One type of such a ranking-loss is the *supremum-loss* in the classical imitation learning setup.

We propose a class of ranking-loss functions $L_k$ that *attempts* to induce a performance gap of $k \in [0, R_{max}]$ for all behavior preferences in the dataset. Formally, this can be implemented with the regression loss:

$$L_k(\mathcal{D}^p; R) = \mathbb{E}_{(\rho^{\pi^i},\rho^{\pi^j})\sim\mathcal{D}^p} \left[ \mathbb{E}_{s,a\sim\rho^{\pi^i}} \left[(R(s,a)-0)^2\right] + \mathbb{E}_{s,a\sim\rho^{\pi^j}} \left[(R(s,a)-k)^2\right] \right]. \tag{3}$$

where $\mathcal{D}^p$ contains behavior pairs $(\rho^{\pi^i}, \rho^{\pi^j})$ with the prespecified ranking $\rho^{\pi^i} \preceq \rho^{\pi^j}$.

The proposed ranking loss allows for learning *bounded rewards with user-defined scale $k$* in the agent and the expert visitations as opposed to prior works in Adversarial Imitation Learning (Ho & Ermon, 2016; Fu et al., 2017; Ghasemipour et al., 2020). Reward scaling has been known to improve learning efficiency in deep RL; a large reward scale can make the optimization landscape less smooth (Henderson et al., 2018; Glorot &

Bengio, 2010) and a small scale might make the action-gap small and increase susceptibility to extrapolation errors (Bellemare et al., 2016). In contrast to the *supremum* loss, $L_k$ can also naturally incorporate rankings provided by additional sources by learning a reward function satisfying all specified pairwise preferences. The following theorem characterizes the equilibrium of the `rank-game` for imitation learning when $L_k$ is used as the ranking-loss.

**Theorem 4.1.** *(Performance of the `rank-game` equilibrium pair) Consider an equilibrium of the imitation* `rank-game` $(\hat{\pi}, \hat{R})$, *such that the ranking loss $L_k$ generalization error is bounded by $2R_{max}^2\epsilon_r$ and the policy is near-optimal with $J(\hat{R}; \hat{\pi}) \geq J(\hat{R}; \pi) - \epsilon_\pi \; \forall \pi$, then at this equilibrium pair under the expert's unknown reward function $R_{gt}$ bounded in $[0, R_{max}^E]$:*

$$\left| J(R_{gt}, \pi^E) - J(R_{gt}, \hat{\pi}) \right| \leq \frac{4R_{max}^E \sqrt{\frac{(1-\gamma)\epsilon_\pi + 4R_{max}\sqrt{\epsilon_r}}{k}}}{1 - \gamma} \tag{4}$$

*If reward is a state-only function and only expert observations are available, the same bound applies to the LfO setting.*

*Proof.* We defer the proof to Appendix A. $\square$

**Comments on Theorem 4.1**: The ranking loss trains the reward function with finite samples using supervised learning. We can quantify $\epsilon_r$, the finite sample generalization error for the reward function, using standard concentration bounds (Shalev-Shwartz & Ben-David, 2014; Hoeffding, 1994) with high probability. We use $\epsilon_\pi$ to denote the policy optimization error from solving the reinforcement learning problem. In Deep Reinforcement Learning, this error can stem as a result of function approximation, biases in value function update, and finite samples. Accounting for this error allows us to bring our analysis closer to the real setting. Note that the performance gap between agent policy and expert policy depends on the scale of the expert reward function $R_{max}^E$. This behavior is expected as the performance gap arising as a result of differences in behaviors/visitations of agent policy and expert policy, can be amplified by the expert's unknown reward scale. We assume realizability i.e the expert reward function lies in the agent

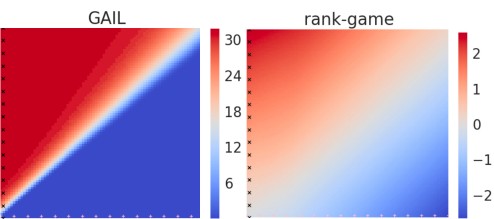

Figure 2: Figure shows learned reward function when agent and expert has a visitation shown by pink and black markers respectively. `rank-game` (auto) results in smooth reward functions more amenable to gradient-based policy optimization compared to GAIL.

reward function class, which ensures that $R_{max}^E \leq R_{max}$. The performance bound in Theorem 4.1 is derived in Appendix A by first proving an intermediate result that demonstrates $\rho^{\hat{\pi}}$ and $\rho^{\pi^E}$ are close in a specific $f$-divergence at the equilibrium, a bound that does not depend on the unknown expert reward scale $R_{max}^E$.

**Theoretical properties:** We now discuss some theoretical properties of $L_k$. Theorem 4.1 shows that `rank-game` has an equilibrium with bounded performance gap with the expert. Second, our derivation for Theorem 4.1 also shows that — an optimization step by the policy player, under a reward function optimized by the reward player, is equivalent to minimizing an $f$-divergence with the expert. Equivalently, at iteration $t$ in Algorithm 1: $\max_{\pi^t} \mathbb{E}_{\rho^{\pi^t}}[R_t^*] - \mathbb{E}_{\rho^{\pi^E}}[R_t^*] = \min_{\pi^t} D_f(\rho^{\pi^t} \| \rho^{\pi^E})$. We characterize and elaborate on the regret of this idealized algorithm in Appendix A. Theorem 4.1 suggests that large values of $k$, upto $R_{max}$, can guarantee the agent's performance is close to the expert. In practice, we observe intermediate values of $k$ also preserve imitation equilibrium optimality with a benefit of promoting sample efficient learning. We attribute this observation to the effect of reward scaling described earlier. We validate this observation further in Appendix D.9. `rank-game` naturally extends to the LfO regime under a state-only reward function where Theorem 4.1 results in a divergence bound between state-visitations of the expert and the agent. A state-only reward function is also a sufficient and necessary condition to ensure that we learn a dynamics-disentangled reward function (Fu et al., 2017).

$L_k$ can incorporate additional preferences that can help learn a regularized/shaped reward function that provides better guidance for policy optimization, reducing the exploration burden and increasing sample efficiency

for IRL. A better-guided policy optimization is also expected to incur a lower $\epsilon_\pi$. However, augmenting the ranking dataset can lead to decrease in the intended performance gap ($k_{eff} < k$) between the agent and the expert (Appendix A). This can loosen the bound in Eq 4 and lead to sub-optimal imitation learning. We hypothesize that given informative preferences, decreased $\epsilon_\pi$ can compensate potentially decreased intended performance gap $k_{eff}$ to ensure near optimal imitation. In our experiments, we observe this hypothesis holds true; we enjoy sample efficiency benefits without losing any asymptotic performance. To leverage these benefits, we present two methods for augmenting the ranking dataset below and defer the implementation details to Appendix B.

### 4.2.1 Generating the Ranking Dataset

**Reward loss w/ automatically generated rankings (auto)**: In this method, we assume access to the behavior-generating trajectories in the ranking dataset. A trajectory $\boldsymbol{\tau}$ is a sequence of states (LfO) given by $[s_0, s_1, ..s_H]$ or state-action pairs (LfD) given by $[s_0, a_0, s_1, a_1..s_H, a_H]$. For each pairwise comparison $\rho_i \preceq \rho_j$ present in the dataset, $L_k$ sets the regression targets for states in $\rho_i$ to be 0 and for states visited by $\rho_j$ to be $k$. Equivalently, we can rewrite minimizing $L_k$ as regressing an input of trajectory $\boldsymbol{\tau}_i$ to vector $\mathbf{0}$, and $\boldsymbol{\tau}_j$ to vector $k\mathbf{1}$ where $\boldsymbol{\tau}_i, \boldsymbol{\tau}_j$ are trajectories that generate the behavior $\rho_i, \rho_j$ respectively. We use the comparison $\rho_i \preceq \rho_j$ to generate additional behavior rankings $\rho_i = \rho_{\lambda_0, ij} \preceq \rho_{\lambda_1, ij} \preceq \rho_{\lambda_2, ij}.. \preceq \rho_{\lambda_P, ij} \preceq \rho_j = \rho_{\lambda_{P+1}, ij}$ where $0 = \lambda_0 < \lambda_1 < \lambda_2 < ... < \lambda_P < 1 = \lambda_{P+1}$. The behavior $\rho_{\lambda_p, ij}$ is obtained by independently sampling the trajectories that generate the behaviors $\rho_i, \rho_j$ and taking convex combinations i.e. $\boldsymbol{\tau}_{\lambda_p, ij} = \lambda_p \boldsymbol{\tau}_i + (1 - \lambda_p)\boldsymbol{\tau}_j$ and their corresponding reward regressions targets are given by $k_p = \lambda_p \mathbf{0} + (1 - \lambda_p)k\mathbf{1}$. The loss function takes the following form:

$$SL_k(\mathcal{D}; R) = \mathbb{E}_{\rho_i, \rho_j \sim \mathcal{D}} \left[ \frac{1}{P+2} \sum_{p=0}^{P+1} \mathbb{E}_{s,a \sim \rho_{\lambda_p, ij}(s,a)} \left[ (R(s,a) - k_p)^2 \right] \right] \tag{5}$$

This form of data augmentation can be interpreted as mixup (Zhang et al., 2017) regularization in the trajectory space. Mixup has been shown to improve generalization and adversarial robustness (Guo et al., 2019; Zhang et al., 2017) by regularizing the first and second order gradients of the parameterized function. Following the general principle of using a smoothed objective with respect to inputs to obtain effective gradient signals, explicit smoothing in the trajectory-space can also help reduce the policy optimization error $\epsilon_\pi$. A didactic example showing rewards learned using this method is shown in Figure 2. In a special case when the expert's unknown reward function is linear in observations, these rankings reflect the true underlying rankings of behaviors.

**Reward loss w/ offline annotated rankings (pref)**: Another way of increasing learning efficiency is augmenting the ranking dataset containing the vanilla ranking ($\{\rho^\pi \preceq \rho^E\} \coloneqq \mathcal{D}^\pi$) with offline annotated rankings ($\mathcal{D}^{offline}$). These rankings may be provided by a human observer or obtained using an offline dataset of behaviors with annotated reward information, similar to the datasets used in offline RL (Fu et al., 2020; Levine et al., 2020). We combine offline rankings by using a weighted loss between $L_k$ for satisfying vanilla rankings ($\rho^\pi \preceq \rho^E$) and offline rankings, grounded by an expert. Providing offline rankings alone that are sufficient to explain the reward function of the expert (Brown et al., 2019) is often a difficult task and the number of offline preferences required depends on the complexity of the environment. In the LfO setting, learning from an expert's state visitation alone can be a hard problem due to exploration requirements (Kidambi et al., 2021). This ranking-loss combines the benefits of using preferences to shape the reward function and guide policy improvement while using the expert to guarantee near-optimal performance. The weighted loss function for this setting takes the following form:

$$L_k(\mathcal{D}^\pi, \mathcal{D}^{offline}; R) = \alpha L_k(\mathcal{D}^\pi; R) + (1 - \alpha) * L_k(\mathcal{D}^{offline}; R) \tag{6}$$

## 4.3 Optimizing the Two-Player General-Sum Ranking Game as a Stackelberg Game

Solving the ranking-game in the Stackelberg setup allows us to propose two different algorithms depending on which agent is set to be the leader and utilize the learning stability and efficiency afforded by the formulation as studied in Rajeswaran et al. (2020); Zheng et al. (2021); Fiez et al. (2019).

**Policy as leader (PAL)**: Choosing policy as the leader implies the following optimization:

$$\max_{\pi} \left\{ J(\hat{R}; \pi) \quad s.t. \quad \hat{R} = \arg\min_{R} L(\mathcal{D}^{\pi}; R) \right\} \tag{7}$$

**Reward as leader (RAL):** Choosing reward as the leader implies the following optimization:

$$\min_{\hat{R}} \left\{ L(\mathcal{D}^{\pi}; \hat{R}) \quad s.t \quad \pi = \arg\max_{\pi} J(\hat{R}; \pi) \right\} \tag{8}$$

We follow the first order gradient approximation for leader's update from previous work (Rajeswaran et al., 2020) to develop practical algorithms. This strategy has been proven to be effective and avoids the computational complexity of calculating the implicit Jacobian term ($d\theta_B^*/d\theta_A$). PAL updates the reward to near convergence on dataset $\mathcal{D}^{\pi}$ ($\mathcal{D}^{\pi}$ contains rankings generated using the current policy agent only $\pi \preceq \pi^E$) and takes a few policy steps. Note that even after the first-order approximation, this optimization strategy differs from GDA as often only a few iterations are used for training the reward even in hyperparameter studies like Orsini et al. (2021). RAL updates the reward conservatively. This is achieved through aggregating the dataset of implicit rankings from all previous policies obtained during training. PAL's strategy of using on-policy data $\mathcal{D}^{\pi}$ for reward training resembles that of methods including GAIL (Ho & Ermon, 2016; Torabi et al., 2018b), $f$-MAX (Ghasemipour et al., 2020), and $f$-IRL (Ni et al., 2020). RAL uses the entire history of agent visitation to update the reward function and resembles methods such as apprenticeship learning and DAC (Abbeel & Ng, 2004; Kostrikov et al., 2018). PAL and RAL bring together two seemingly different algorithm classes under a unified Stackelberg game viewpoint.

## 5  Experimental Results

We compare `rank-game` against state-of-the-art LfO and LfD approaches on MuJoCo benchmarks having continuous state and action spaces. The LfO setting is more challenging since no actions are available, and is a crucial imitation learning problem that can be used in cases where action modalities differ between the expert and the agent, such as in robot learning. We focus on the LfO setting in this section and defer the LfD experiments to Appendix D.2. We denote the imitation learning algorithms that use the proposed ranking-loss $L_k$ from Section 4.2 as RANK-{PAL, RAL}. We refer to the `rank-game` variants which use automatically generated rankings and offline preferences as (auto) and (pref) respectively following Section 4.2. In all our methods, we rely on an off-policy model-free algorithm, Soft Actor-Critic (SAC) (Haarnoja et al., 2018), for updating the policy agent (in step 5 of Algorithm 1).

We design experiments to answer the following questions:

1. *Asymptotic Performance and Sample Efficiency*: Is our method able to achieve near-expert performance given a limited number (one) of expert observations? Can our method learn using fewer environment interactions than prior state-of-the-art imitation learning (LfO) methods?

2. *Utility of preferences for imitation learning*: Current LfO methods struggle to solve a number of complex manipulation tasks with sparse success signals. Can we leverage offline annotated preferences through `rank-game` in such environments to achieve near-expert performance?

3. *Choosing between PAL and RAL methods*: Can we characterize the benefits and pitfalls of each method, and determine when one method is preferable over the other?

4. *Ablations for the method components*: Can we establish the importance of hyperparameters and design decisions in our experiments?

**Baselines:** We compare RANK-PAL and RANK-RAL against 6 representative LfO approaches that covers a spectrum of on-policy and off-policy model-free methods from prior work: GAIfO (Torabi et al., 2018b; Ho & Ermon, 2016), DACfO  (Kostrikov et al., 2018), BCO (Torabi et al., 2018a), $f$-IRL (Ni et al., 2020) and recently proposed OPOLO (Zhu et al., 2020b) and IQLearn (Garg et al., 2021). We do not assume

access to expert actions in this setting. Our LfD experiments compare to the IQLearn (Garg et al., 2021), DAC (Kostrikov et al., 2018) and BC baselines. Detailed description about the environments and baselines can be found in Appendix C.

## 5.1 Asymptotic Performance and Sample Efficiency

In this section, we compare RANK-PAL(auto) and RANK-RAL(auto) to baselines on a set of MuJoCo locomotion tasks of varying complexities: `Swimmer-v2`, `Hopper-v2`, `HalfCheetah-v2`, `Walker2d-v2`, `Ant-v2` and `Humanoid-v2`. In this experiment, we provide one expert trajectory for all methods and do not assume access to any offline annotated rankings.

| Env | Hopper | HalfCheetah | Walker | Ant | Humanoid |
|---|---|---|---|---|---|
| BCO | 20.10±2.15 | 5.12±3.82 | 4.00±1.25 | 12.80±1.26 | 3.90±1.24 |
| GaIFO | 81.13± 9.99 | 13.54±7.24 | 83.83±2.55 | 20.10±24.41 | 3.93±1.81 |
| DACfO | 94.73±3.63 | 85.03±5.09 | 54.70±44.64 | 86.45±1.67 | 19.31±32.19 |
| $f$-IRL | 97.45± 0.61 | 96.06±4.63 | **101.16±1.25** | 71.18±19.80 | 77.93±6.372 |
| OPOLO | 89.56±5.46 | 88.92±3.20 | 79.19±24.35 | 93.37± 3.78 | 24.87±17.04 |
| RANK-PAL(ours) | 87.14± 16.14 | 94.05±3.59 | 93.88±0.72 | **98.93±1.83** | **96.84±3.28** |
| RANK-RAL(ours) | **99.34±0.20** | **101.14±7.45** | 93.24±1.25 | 93.21±2.98 | 94.45±4.13 |
| Expert | 100.00± 0 | 100.00± 0 | 100.00± 0 | 100.00± 0 | 100.00± 0 |
| $(|\mathcal{S}|, |\mathcal{A}|)$ | (11, 3) | (17, 6) | (17, 6) | (111, 8) | (376, 17) |

Table 2: Asymptotic normalized performance of LfO methods at 2 million timesteps on MuJoCo locomotion tasks. The standard deviation is calculated with 5 different runs each averaging over 10 trajectory returns. For unnormalized score and more details, check Appendix D. We omit IQlearn due to poor performance.

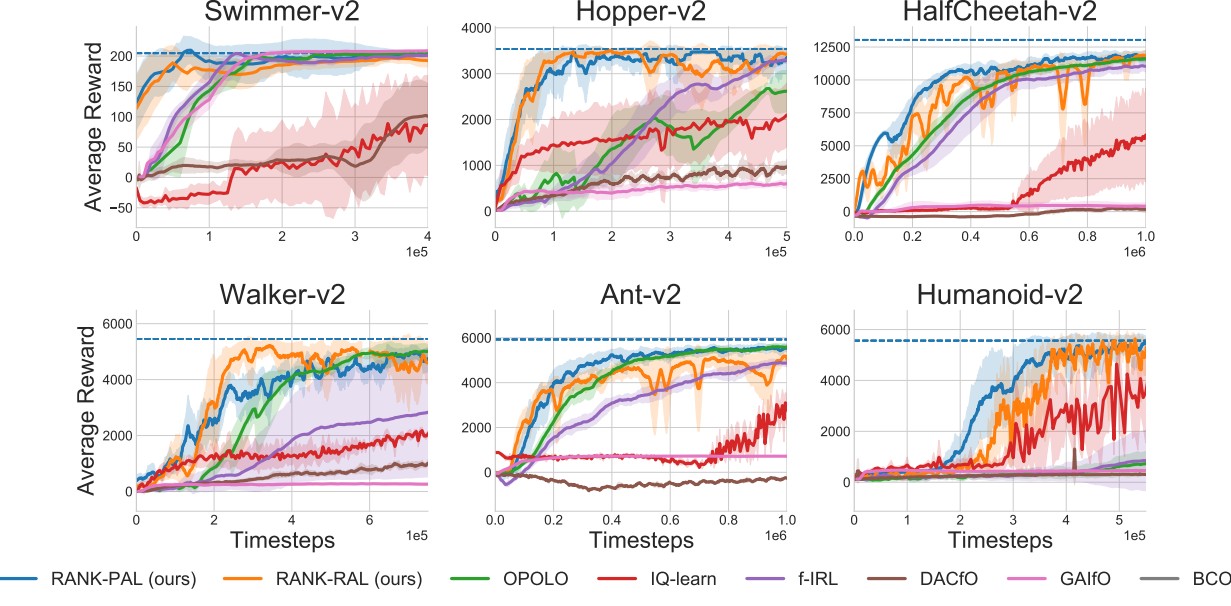

Figure 3: Comparison of performance on OpenAI gym benchmark tasks. The shaded region represents standard deviation across 5 random runs. RANK-PAL and RANK-RAL substantially outperform the baselines in sample efficiency. Complete set of results can be found in Appendix D.1

**Asymptotic Performance**: Table 2 shows that both `rank-game` methods are able to reach near-expert asymptotic performance with a single expert trajectory. BCO shows poor performance which can be attributed

to the compounding error problem arising from its behavior cloning strategy. GAIfO and DACfO use GDA for optimization with a supremum loss and show high variance in their asymptotic performance whereas `rank-game` methods are more stable and low-variance.

**Sample Efficiency**: Figure 3 shows that RANK-RAL and RANK-PAL are among the most sample efficient methods for the LfO setting, outperforming the recent state-of-the-art method OPOLO (Zhu et al., 2020b) by a significant margin. We notice that IQLearn fails to learn in the LfO setting. This experiment demonstrates the benefit of the combined improvements of the proposed ranking-loss with automatically generated rankings. Our method is also simpler to implement than OPOLO, as we require fewer lines of code changes on top of SAC and need to maintain fewer parameterized networks compared to OPOLO which requires an additional inverse action model to regularize learning.

## 5.2 Utility of Preferences in Imitation

Our experiments on complex manipulation environments—door opening with a parallel-jaw gripper (Zhu et al., 2020a) and pen manipulation with a dexterous adroit hand (Rajeswaran et al., 2017) – reveal that none of the prior LfO methods are able to imitate the expert even under increasing amounts of expert data. This failure of LfO methods can be potentially attributed to the exploration requirements of LfO compared to LfD (Kidambi et al., 2021), coupled with the sparse successes encountered in these tasks, leading to poorly guided policy gradients.

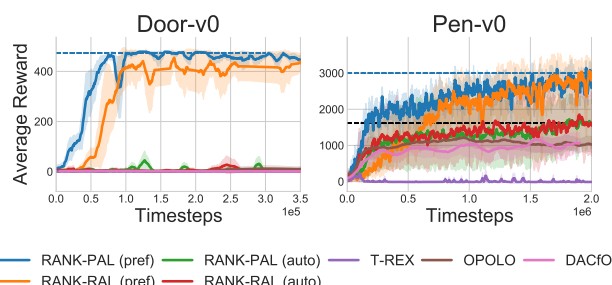

Figure 4: Offline annotated preferences can help solve LfO tasks in the complex manipulation environments Pen-v0 and Door, whereas prior LfO methods fail. Black dotted line shows asymptotic performance of RANK-PAL (auto) method.

In these experiments, we show that `rank-game` can incorporate additional information in the form of offline annotated rankings to guide the agent in solving such tasks. These offline rankings are obtained by uniformly sampling a small set of trajectories (10) from the replay buffer of SAC (Haarnoja et al., 2018) labeled with a ground truth reward function. We use a weighted ranking loss (pref) from Section 4.2.

Figure 4 shows that RANK-PAL/RAL(pref) method leveraging offline ranking is the only method that can solve these tasks, whereas prior LfO methods and RANK-PAL/RAL(auto) with automatically generated rankings struggle even after a large amount of training. We also point out that T-REX, an offline method that learns using the preferences grounded by expert is unable to achieve near-expert performance, thereby highlighting the benefits of learning online with expert demonstrations alongside a set of offline preferences.

## 5.3 Comparing PAL and RAL

PAL uses the agent's current visitation for reward learning, whereas RAL learns a reward consistent with all rankings arising from the history of the agent's visitation. These properties can present certain benefits depending on the task setting.

To test the potential benefits of PAL and RAL, we consider two non-stationary imitation learning problems, similar to (Rajeswaran et al., 2017) – one in which the expert changes it's intent and the other where dynamics of the environment change during training in the Hopper-v2 locomotion task. For changing intent, we present a new set of demonstrations where the hopper agent hops backwards rather than forward. For changing environment dynamics, we increase the mass of the hopper agent by a factor of 1.2. Changes are introduced at 1e5 time steps during training at which point we notice a sudden performance drop.

In Figure 5 (left), we notice that PAL adapts faster to intent changes, whereas RAL needs to unlearn the rankings obtained from the agent's history and takes longer to adapt. Figure 5 (right) shows that RAL adapts faster to the changing dynamics of the system, as it has already learned a good global

notion of the dynamics-disentangled reward function in the LfO setting, whereas PAL only has a local understanding of reward as a result of using ranking obtained only from the agent's current visitation.

**Ablation of Method Components:** Appendix D contains eight additional experiments to study the importance of hyperparameters and design decisions. Our ablations validate the importance of using automatically generated rankings, the benefit of ranking loss over *supremum* loss, and sensitivity to hyperparameters like the intended performance gap $k$, policy iterations, and the reward regularizer. We find that key improvements in learning efficiency are driven by using the proposed ranking loss, controlling the reward range, and the reward/policy update frequency in the Stackelberg framework. In Figure 6, we also analyze the performance of `rank-game` with a varying number of expert trajectories and its robustness to noisy offline-annotated preferences. We find `rank-game` to consistently outperform baselines with a varying number of expert trajectories. On Door manipulation task `rank-game` is robust to 60 percent noise in the offline-annotated preferences. Experiments on more environments and additional details can be found in Appendix D.

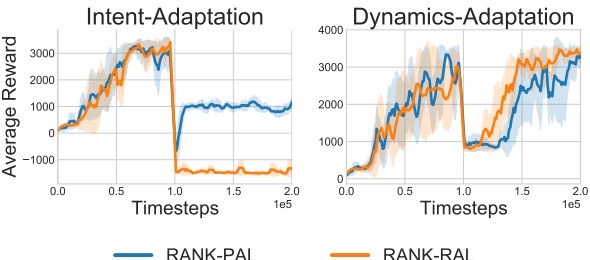

Figure 5: We compare the relative strengths of PAL and RAL. Left plot shows a comparison when the goal is changed, and right plot shows a comparison when dynamics of the environment is changed. These changes occur at 1e5 timesteps into training. PAL adapts faster to changing intent and RAL adapts faster to changing dynamics.

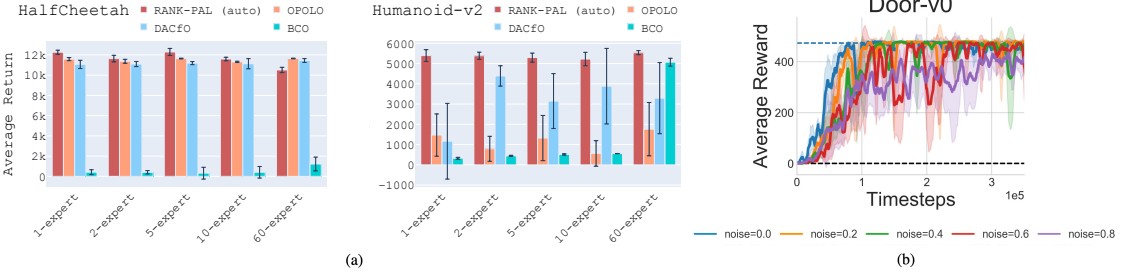

Figure 6: (a) RANK-PAL outperforms other methods with varying number of expert trajectories. Error bars denote standard deviation. (b) On Door-v0 environment, RANK-PAL(pref) is robust to at least 60 percent noisy preferences.

## 6 Conclusion

In this work, we present a new framework for imitation learning that treats imitation as a two-player ranking-game between a policy and a reward function. Unlike prior works in imitation learning, the ranking game allows incorporation of rankings over suboptimal behaviors to aid policy learning. We instantiate the ranking game by proposing a novel ranking loss which guarantees agent's performance to be close to expert for imitation learning. Our experiments on simulated MuJoCo tasks reveal that utilizing additional ranking through our proposed ranking loss leads to improved sample efficiency for imitation learning, outperforming prior methods by a significant margin and solving some tasks which were unsolvable by previous LfO methods.

**Limitations and Negative Societal Impacts:** Preferences obtained in the real world are usually noisy (Kwon et al., 2020; Jeon et al., 2020; Bıyık et al., 2021) and one limitation of `rank-game` is that it does not suggest a way to handle noisy preferences. Second, `rank-game` proposes modifications to learn a reward function amenable to policy optimization but these hyperparameters are set manually. Future work can explore methods to automate learning such reward functions. Third, despite learning effective policies we observed that we do not learn reusable robust reward functions (Ni et al., 2020). Negative Societal Impact: Imitation learning can cause harm if given demonstrations of harmful behaviors, either accidentally or purposefully. Furthermore, even when given high-quality demonstrations of desirable behaviors, our algorithm does not provide guarantees of performance, and thus could cause harm if used in high-stakes domains without sufficient safety checks on learned behaviors.

## Acknowledgements

We thank Jordan Schneider, Wenxuan Zhou, Caleb Chuck, and Amy Zhang for valuable feedback on the paper. This work has taken place in the Personal Autonomous Robotics Lab (PeARL) at The University of Texas at Austin. PeARL research is supported in part by the NSF (IIS-1724157, IIS-1749204, IIS-1925082), AFOSR (FA9550-20-1-0077), and ARO (78372-CS). This research was also sponsored by the Army Research Office under Cooperative Agreement Number W911NF-19-2-0333. The views and conclusions contained in this document are those of the authors and should not be interpreted as representing the official policies, either expressed or implied, of the Army Research Office or the U.S. Government. The U.S. Government is authorized to reproduce and distribute reprints for Government purposes notwithstanding any copyright notation herein.

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

# A  Theory

We aim to show that `rank-game` has an equilibrium that bounds the $f$-divergence between the agent and the expert (Theorem A.1) in the imitation learning setting. For imitation learning, we have the vanilla implicit ranking $\rho^{agent} \preceq \rho^E$, between the behavior of agent and the expert. Later, we show that, the bounded $f$-divergence can be used to bound the performance gap with the expert under the expert's unknown reward function using a solution to Vajda's tight lower bound (Corollary A.1.1). Our proof proceeds by first showing that minimizing the empirical ranking loss produces a reward function that is close to the reward function obtained by the true ranking loss. Then, we show that even under the presence of policy optimization errors maximizing the obtained reward function will lead to a bounded $f$-divergence with the expert.

**Theorem A.1.** *(Performance of the* `rank-game` *equilibrium pair) Consider an equilibrium of the imitation* `rank-game` $(\hat{\pi}, \hat{R})$, *such that* $\hat{R}$ *minimizes the empirical ranking-loss for dataset* $D^{\hat{\pi}} = \{(\rho^{\hat{\pi}}, \rho^E)\}$ *and the ranking-loss generalization error is bounded by* $\epsilon'_r = 2R_{max}^2\epsilon_r$, *and the policy* $\hat{\pi}$ *has bounded suboptimality with* $J(\hat{R}; \hat{\pi}) \geq J(\hat{R}; \pi') - \epsilon_\pi \ \forall \pi'$, *then we have that at this equilibrium pair:*

$$D_f\left(\rho^{\hat{\pi}}(s,a) || \rho^E(s,a)\right) \leq \frac{(1-\gamma)\epsilon_\pi + 4R_{max}\sqrt{2\epsilon_r}}{k} \tag{9}$$

*where* $D_f$ *is an* $f$-divergence *with the generator function* $f(x) = \frac{1-x}{1+x}$ *(Rényi, 1961; Ali & Silvey, 1966; Csiszár, 1967; Liese & Vajda, 2006).*

*Proof.* Previous works (Xu et al., 2021; Swamy et al., 2021) characterize the equilibrium in imitation learning based on the *supremum* ranking loss/min-max adversarial setting under no error assumption. In this section, we consider the ranking loss function $L_k$ and derive the equilibrium for the `rank-game` in presence of reward learning and policy optimization errors. $L_k$ attempts to explain the rankings between the agent and the expert using their state-action visitations $\mathcal{D}^\pi = \{\rho^\pi(s,a), \rho^E(s,a)\}$ respectively, by attempting to induce a performance gap of $k$. With this dataset $\mathcal{D}^\pi$, $L_k$ regresses the return of state or state-action pairs in the expert's visitation to a scalar $k$ and the agent's visitation to a value of 0. Thus, we have:

$$L_k(\mathcal{D}; R) = \mathbb{E}_{\rho^E(s,a)}\left[(R(s,a) - k)^2\right] + \mathbb{E}_{\rho^\pi(s,a)}\left[(R(s,a) - 0)^2\right] \tag{10}$$

The above ranking loss is minimized ($\nabla L_k = 0$) pointwise when

$$R^*(s,a) = \frac{k(\rho^E(s,a))}{\rho^E(s,a) + \rho^\pi(s,a)} \tag{11}$$

In practice, we have finite samples from both the expert visitation distribution and the agent distribution so we minimize the following empirical ranking loss $\hat{L}_k(\mathcal{D}; R)$:

$$\hat{L}_k(\mathcal{D}; R) = \frac{\sum_{s,a \in \hat{\rho}^E}[(R(s,a) - k)^2]}{|\hat{\rho}^E|} + \frac{\sum_{s,a \in \hat{\rho}^\pi}[(R(s,a) - 0)^2]}{|\hat{\rho}^\pi|} \tag{12}$$

where $\hat{\rho}^E$ and $\hat{\rho}^\pi$ are empirical state-action visitations respectively.

**From empirical loss function to reward optimality:** Since the reward function is trained with supervised learning, we can quantify the sample error in minimizing the empirical loss using concentration bounds (Shalev-Shwartz & Ben-David, 2014) up to a constant with high probability. Since $0 < R(s,a) < R_{max}$ With high probability,

$$\forall R, \quad |L_k(\mathcal{D}; R) - \hat{L}_k(\mathcal{D}; R)| \leq 2R_{max}^2\epsilon_r \tag{13}$$

where $\epsilon_r$ is the statistical estimation error that can be bounded using concentration bounds such as Hoeffding's. Let $R^*$ belong to the optimal solution for $L_k(\mathcal{D}; R)$ and $\hat{R}^*$ belong to the optimal minimizing solution for $\hat{L}_k(\mathcal{D}; R)$. Therefore, we have that,

$$\forall R, \quad \hat{L}_k(\mathcal{D}; \hat{R}^*) \leq \hat{L}_k(\mathcal{D}; R) \tag{14}$$

Using Eq 13 and Eq 14, we have

$$\forall R, \quad \hat{L}_k(\mathcal{D}; \hat{R}^*) \leq \hat{L}_k(\mathcal{D}; R) \tag{15}$$

$$\leq L_k(\mathcal{D}; R) + 2R_{max}^2\epsilon_r \tag{16}$$

$$\leq L_k(\mathcal{D}; R^*) + 2R_{max}^2\epsilon_r \tag{17}$$

and similarly

$$\forall R, \quad L_k(\mathcal{D}; R^*) \leq L_k(\mathcal{D}; R) \tag{18}$$

$$\leq \hat{L}_k(\mathcal{D}; R) + 2R_{max}^2 \epsilon_r \tag{19}$$

$$\leq \hat{L}_k(\mathcal{D}; \hat{R}^*) + 2R_{max}^2 \epsilon_r \tag{20}$$

Eq 17 and Eq 20 implies that $L_k(\mathcal{D}; R^*)$ and $\hat{L}_k(\mathcal{D}; \hat{R}^*)$ are bounded with high probability. i.e

$$|L_k(\mathcal{D}; R^*) - \hat{L}_k(\mathcal{D}; \hat{R}^*)| \leq 2R_{max}^2 \epsilon_r \tag{21}$$

We will use Eq 21 to show that indeed $\hat{R}^*$ has a bounded loss compared to $R^*$.

$$\hat{L}_k(\mathcal{D}; \hat{R}^*) - L_k(\mathcal{D}; R^*) \leq 2R_{max}^2 \epsilon_r \tag{22}$$

$$L_k(\mathcal{D}; \hat{R}^*) - 2R_{max}^2 - L_k(\mathcal{D}; R^*)\epsilon_r \leq 2R_{max}^2 \epsilon_r \tag{23}$$

$$L_k(\mathcal{D}; \hat{R}^*) - L_k(\mathcal{D}; R^*) \leq 4R_{max}^2 \epsilon_r \tag{24}$$

We consider the tabular MDP setting and overload R to denote a vector of reward values for the entire state-action space of size $|\mathcal{S} \times \mathcal{A}|$. Using the Taylor series expansion for loss function $L_k$, we have:

$$L_k(\mathcal{D}; \hat{R}^*) - L_k(\mathcal{D}; R^*) \leq 4R_{max}^2 \epsilon_r \tag{25}$$

$$L_k(\mathcal{D}; R^*) + \langle \nabla_{R^*} L_k(\mathcal{D}; R^*), \hat{R}^* - R^* \rangle$$
$$+0.5(\hat{R}^* - R^*)^T H(\hat{R}^* - R^*) - L_k(\mathcal{D}; R^*) \leq 4R_{max}^2 \epsilon_r \tag{26}$$

$$(\hat{R}^* - R^*)^T H(\hat{R}^* - R^*) \leq 8R_{max}^2 \epsilon_r \tag{27}$$

where $H$ denotes the hessian for the loss function w.r.t $R$ and is given by $H = P^{\rho^\pi} + P^{\rho^E}$ where $P^\rho$ is a matrix of size $|\mathcal{S} \times \mathcal{A}| \times |\mathcal{S} \times \mathcal{A}|$ with $\rho$ vector of visitations as its diagonal and zero elsewhere.

$$(\hat{R}^* - R^*)^T H(\hat{R}^* - R^*) \leq 8R_{max}^2 \epsilon_r \tag{28}$$

$$\mathbb{E}_{s \sim \rho^\pi}\left[(\hat{R}^*(s,a) - R^*(s,a))^2\right] + \mathbb{E}_{s \sim \rho^E}\left[(\hat{R}^*(s,a) - R^*(s,a))^2\right] \leq 8R_{max}^2 \epsilon_r \tag{29}$$

Since both terms in the LHS are positive we have $\mathbb{E}_{s,a \sim \rho^\pi}\left[(\hat{R}^*(s,a) - R^*(s,a))^2\right] \leq 8R_{max}^2 \epsilon_r$ and $\mathbb{E}_{s,a \sim \rho^E}\left[(\hat{R}^*(s,a) - R^*(s,a))^2\right] \leq 8R_{max}^2 \epsilon_r$. Applying Jensen's inequality, we further have $(\mathbb{E}_{s,a \sim \rho^\pi}\left[\hat{R}^*(s,a) - R^*(s,a)\right])^2 \leq 8R_{max}^2 \epsilon_r$ and $(\mathbb{E}_{s,a \sim \rho^E}\left[\hat{R}^*(s,a) - R^*(s,a)\right])^2 \leq 8R_{max}^2 \epsilon_r$. Hence,

$$\left|\mathbb{E}_{s,a \sim \rho^\pi}\left[\hat{R}^*(s,a) - R^*(s,a)\right]\right| \leq R_{max}\sqrt{8\epsilon_r} \quad \text{, and} \tag{30}$$

$$\left|\mathbb{E}_{s,a \sim \rho^E}\left[\hat{R}^*(s,a) - R^*(s,a)\right]\right| \leq R_{max}\sqrt{8\epsilon_r} \tag{31}$$

At this point we have bounded the expected difference between the reward functions obtained from the empirical ranking loss and the true ranking loss. We will now characterize the equilibrium obtained by learning a policy on the reward function $\hat{R}^*$ that is optimal under the empirical ranking loss. Under a policy optimization error of $\epsilon_\pi$ we have:

$$J(\hat{R}^*; \hat{\pi}) \geq J(\hat{R}^*; \pi') - \epsilon_\pi \ \forall \pi' \in \Pi \tag{32}$$

where $J(R; \pi)$ denotes the performance of policy $\pi$ under reward function $R$.

Taking $\pi' = \pi^E$, we can reduce the above expression as follows:

$$J(\hat{R}^*, \pi^E) - J(\hat{R}^*, \hat{\pi}) \leq \epsilon_\pi \tag{33}$$

$$\frac{1}{1-\gamma}\left[\mathbb{E}_{\rho^E(s,a)}\left[\hat{R}^*(s,a)\right] - \mathbb{E}_{\rho^\pi(s,a)}\left[\hat{R}^*(s,a)\right]\right] \leq \epsilon_\pi \tag{34}$$

Using Eq 30 and Eq 31 we can lower bound $\mathbb{E}_{\rho^E(s,a)}\left[\hat{R}^*(s,a)\right] - \mathbb{E}_{\rho^\pi(s,a)}\left[\hat{R}^*(s,a)\right]$ as follows:

$$\mathbb{E}_{\rho^E(s,a)}\left[\hat{R}^*(s,a)\right] \geq \mathbb{E}_{\rho^E(s,a)}[R^*(s,a)] - R_{max}\sqrt{8\epsilon_r} \tag{35}$$

$$\mathbb{E}_{\rho^\pi(s,a)}\left[\hat{R}^*(s,a)\right] \leq \mathbb{E}_{\rho^\pi(s,a)}[R^*(s,a)] + R_{max}\sqrt{8\epsilon_r} \tag{36}$$

where $R^*(s,a)$ is given by Equation 11.

Subtracting Equation 36 from Equation 35, we have

$$\mathbb{E}_{\rho^E(s,a)}\left[\hat{R}^*(s,a)\right] - \mathbb{E}_{\rho^\pi(s,a)}\left[\hat{R}^*(s,a)\right] \geq \mathbb{E}_{\rho^E(s,a)}[R^*(s,a)] - \mathbb{E}_{\rho^\pi(s,a)}[R^*(s,a)] - 2R_{max}\sqrt{8\epsilon_r} \tag{37}$$

Plugging in the lower bound from Equation 37 in Equation 34 we have:

$$\frac{1}{1-\gamma}[\mathbb{E}_{\rho^E(s,a)}[R^*(s,a)] - \mathbb{E}_{\rho^\pi(s,a)}[R^*(s,a)] - 2R_{max}\sqrt{8\epsilon_r}] \leq \epsilon_\pi \tag{38}$$

Replacing $R^*$ using Equation 11 we get,

$$\frac{1}{1-\gamma}\left[\mathbb{E}_{\rho^E(s,a)}\left[\frac{k(\rho^E(s,a))}{\rho^E(s,a) + \rho^\pi(s,a)}\right] - \mathbb{E}_{\rho^\pi(s,a)}\left[\frac{k(\rho^E(s,a))}{\rho^E(s,a) + \rho^\pi(s,a)}\right] - 2R_{max}\sqrt{8\epsilon_r}\right] \leq \epsilon_\pi \tag{39}$$

$$\mathbb{E}_{\rho^E(s,a)}\left[\frac{k(\rho^E(s,a))}{\rho^E(s,a) + \rho^\pi(s,a)}\right] - \mathbb{E}_{\rho^\pi(s,a)}\left[\frac{k(\rho^E(s,a))}{\rho^E(s,a) + \rho^\pi(s,a)}\right] \leq (1-\gamma)\epsilon_\pi + 2R_{max}\sqrt{8\epsilon_r} \tag{40}$$

$$\mathbb{E}_{\rho^E(s,a)}\left[\frac{(\rho^E(s,a) - \rho^\pi(s,a))}{\rho^E(s,a) + \rho^\pi(s,a)}\right] \leq \frac{(1-\gamma)\epsilon_\pi + 2R_{max}\sqrt{8\epsilon_r}}{k} \tag{41}$$

The convex function $f(x) = \frac{1-x}{1+x}$ in $\mathbb{R}^+$ defines an $f$-divergence. Under this generator function, the LHS of Equation 41 defines an $f$-divergence between the state-visitations of the agent $\rho^\pi(s,a)$ and the expert $\rho^E(s,a)$. Hence, we have that

$$D_f[\rho^\pi(s,a), \rho^E(s,a)] \leq \frac{(1-\gamma)\epsilon_\pi + 4R_{max}\sqrt{2\epsilon_r}}{k} \tag{42}$$

This bound shows that the equilibrium of the ranking game is a near-optimal imitation learning solution when ranking loss target $k$ trades off effectively with the policy optimization error $\epsilon_\pi$ and reward generalization error $\epsilon_r$. We note that, since $k \leq R_{max}$ we can get the tightest bound when $k = R_{max}$. Now, in imitation learning both $k$ and $R_{max}$ are tunable hyperparameters. We vary $k$ while keeping $k = R_{max}$ and observe in appendix D.9 that this hyperparameter can significantly affect learning performance. $\qquad\square$

**Corollary A.1.1.** *(From $f$-divergence to performance gap) For the equilibrium of the* `rank-game` *$(\hat{\pi}, \hat{R})$ as described in Theorem A.1, we have that the performance gap of the expert policy with $\hat{\pi}$ is bounded under the unknown expert's reward function $(r_{gt})$ bounded in $[0, R_{max}^E]$ as follows:*

$$|J(\pi^E, r_{gt}) - J(\hat{\pi}, r_{gt})| \leq \frac{4R_{max}^E\sqrt{\frac{(1-\gamma)\epsilon_\pi + 4R_{max}\sqrt{2\epsilon_r}}{k}}}{1-\gamma} \tag{43}$$

*Proof.* In Theorem A.1, we show that the equilibrium of rank-game ensures that.the $f$-divergence of expert visitation and agent visitation is bounded with the generator function $f = \frac{1-x}{1+x}$. First we attempt to find a tight lower bound of our $f$-divergence in terms of the total variational distance between the two distributions. Such a bound has been discussed in previous literature for the usual $f$-divergences like KL, Hellinger and $\chi^2$. This problem of finding a tight lower bound in terms of variational distance for a general $f$-divergence was introduced in Vajda (1970) and referred to as Vajda's tight lower bound and a solution for arbitrary $f$-divergences was proposed in Gilardoni (2006). The $f$-divergence with generator function $f = \frac{1-x}{1+x}$ satisfies that $f(t) = tf(\frac{1}{t}) + 2f'(1)(t-1)$ and hence the total variational bound for this $f$ divergence takes the form

$D_f \geq \frac{2-D_{TV}}{2} f(\frac{2+D_{TV}}{2-D_{TV}}) - f'(1)D_{TV}$. Plugging in the function definition $f = \frac{1-x}{1+x}$ the inequality simplifies to:

$$D_f(\rho^\pi(s,a)\|\rho^E(s,a)) \geq \frac{(D_{TV}(\rho^\pi(s,a)\|\rho^E(s,a))^2}{4} \tag{44}$$

We also note an upper bound for this $f$-divergence in TV distance, sandwiching this particular $f$-divergence with TV bounds:

$$D_f(\rho^\pi(s,a)\|\rho^E(s,a)) = \mathbb{E}_{\rho^E(s,a)}\left[\frac{\rho^E(s,a)}{\rho^E(s,a) + \rho^\pi(s,a)}\right] - \mathbb{E}_{\rho^\pi(s,a)}\left[\frac{\rho^E(s,a)}{\rho^E(s) + \rho^\pi(s,a)}\right] \tag{45}$$

$$\leq \sum_{s,a \in \mathcal{S} \times \mathcal{A}} |\rho^E(s,a) - \rho^\pi(s,a)| \left|\frac{\rho^E(s,a)}{\rho^E(s,a) + \rho^\pi(s,a)}\right| \tag{46}$$

$$\leq D_{TV}(\rho^\pi(s,a)\|\rho^E(s,a)) \tag{47}$$

So,

$$D_{TV}(\rho^\pi(s,a)\|\rho^E(s,a)) \geq D_f(\rho^\pi(s,a)\|\rho^E(s,a)) \geq \frac{(D_{TV}(\rho^\pi(s,a)\|\rho^E(s,a))^2}{4} \tag{48}$$

Therefore from Eq 42 we have that,

$$D_{TV}(\rho^\pi(s,a)\|\rho^E(s,a)) \leq 2\sqrt{\frac{(1-\gamma)\epsilon_\pi + 4R_{max}\sqrt{2\epsilon_r}}{k}} \tag{49}$$

For any policy $\pi$, and experts unknown reward function $r_{gt}$, $J(\pi, r) = \frac{1}{1-\gamma}[\mathbb{E}_{s,a\sim\rho^\pi}[r(s,a)]]$. Therefore,

$$|J(\pi^E, r_{gt}) - J(\pi, r_{gt})| = \left|\frac{1}{1-\gamma}[\mathbb{E}_{s,a\sim\rho^E}[r_{gt}(s,a)]] - \frac{1}{1-\gamma}[\mathbb{E}_{s,a\sim\rho^\pi}[r_{gt}(s,a)]]\right| \quad \forall\pi \tag{50}$$

$$= \frac{1}{1-\gamma}\left|\sum_{s,a\in\mathcal{S}\times\mathcal{A}} |(\rho^E - \rho^\pi)r_{gt}(s,a)|\right| \tag{51}$$

$$\leq \frac{R_{max}^E}{1-\gamma}\sum_{s,a\in\mathcal{S}\times\mathcal{A}} |(\rho^E - \rho^\pi)| \tag{52}$$

$$\leq \frac{2R_{max}^E}{1-\gamma} D_{TV}(\rho_E, \rho_\pi) \tag{53}$$

$$\tag{54}$$

where $R_{max}^E$ is the upper bound for the expert's reward function. Under a worst case expert reward function which assigns finite reward values to the expert's visitation and $-\infty$ outside the visitation, even a small mistake (visiting any state outside the expert's visitation) by the policy can result in an infinite performance gap between expert and the agent. Thus, this parameter is decided by the expert and is not in control of the learning agent.

From Eq 49 and Eq 53 we have

$$|J(\pi^E, r_{gt}) - J(\hat\pi, r_{gt})| \leq \frac{4R_{max}^E \sqrt{\frac{(1-\gamma)\epsilon_\pi + 4R_{max}\sqrt{2\epsilon_r}}{k}}}{1-\gamma} \tag{55}$$

$\square$

**Lemma A.2.** *(Regret bound under finite data assumptions) Let $\hat{M}_t$ denote the approximate transition model under the collected dataset of transitions until iteration $t$. Assume that the ground truth model $M$ and the reward function are realizable. Under these assumptions the regret of* `rank-game` *at $t^{th}$ iteration:*

$$V_M^{\pi^E} - V_M^{\pi^t} \leq \frac{2\gamma\epsilon_m^{\pi^t}R_{max}}{(1-\gamma)^2} + \frac{4R_{max}}{1-\gamma}\sqrt{D_f(\rho_{\hat{M}_t}^{\pi^E}\|\rho_M^{\pi^E})} + \frac{2\epsilon_{stat} + 4R_{max}\sqrt{\epsilon_r}}{k} \tag{56}$$

where $V_M^\pi$ denotes the performance of policy $\pi$ under transition dynamics $M$, $\epsilon_m^{\pi^t}$ is expected model error under policy $\pi^t$'s visitation, $\rho_M^\pi$ is the visitation of policy $\pi$ in transition dynamics $M$ and $\epsilon_{stat}$ is the statistical error due to finite expert samples.

*Proof.* We use $M$ to denote the ground truth model and $\hat{M}_t$ to denote the approximate transition model with collected data until the $t^{th}$ iteration of `rank-game`. We are interested in solving the following optimization problem under finite data assumptions:

$$\max_\pi \mathbb{E}_{s,a\sim\rho_{\hat{M}_t}^\pi}\left[\hat{f}_\pi^*(s,a)\right] - \frac{\sum_{s,a\in\hat{\rho}^E}[\hat{f}_\pi^*(s,a)]}{|\hat{\rho}^E|} \quad s.t \quad \hat{f}_\pi^* = \arg\min_f(\hat{L}_k(f; D_{\hat{M}_t}^\pi)) \tag{57}$$

where $\hat{\rho}^E$ is the empirical distribution generated from finite expert samples and $D_{\hat{M}_t}^\pi = \{(\hat{\rho}_{\hat{M}_t}^\pi, \hat{\rho}_M^E)\}$. Using standard concentration bounds such as Hoeffding's (Hoeffding, 1994), we can bound the empirical estimate with true estimate $\forall\pi$ with high probability:

$$\left|\frac{\sum_{s,a\in\hat{\rho}^E}[f_\pi^*(s,a)]}{|\hat{\rho}^E|} - \mathbb{E}_{s\sim\rho_M^E}[f_\pi^*(s,a)]\right| \leq \epsilon_{stat} \tag{58}$$

Using the concentration bounds and the fact that $\pi^t$ is the solution that maximizes the optimization problem Eq 57 at $t$-iteration,

$$\mathbb{E}_{s,a\sim\rho_{\hat{M}_t}^{\pi^t}}\left[\hat{f}_{\pi^t}^*(s,a)\right] - \frac{\sum_{s,a\in\hat{\rho}^E}[\hat{f}_{\pi^t}^*(s,a)]}{|\hat{\rho}^E|} \geq \mathbb{E}_{s,a\sim\rho_{\hat{M}_t}^{\pi^E}}\left[\hat{f}_{\pi^E}^*(s,a)\right] - \frac{\sum_{s,a\in\hat{\rho}^E}[\hat{f}_{\pi^E}^*(s,a)]}{|\hat{\rho}^E|} \tag{59}$$

$$\geq \mathbb{E}_{s,a\sim\rho_{\hat{M}_t}^{\pi^E}}\left[\hat{f}_{\pi^E}^*(s,a)\right] - \mathbb{E}_{s,a\sim\rho_M^E}\left[\hat{f}_{\pi^E}^*(s,a)\right] - \epsilon_{stat} \tag{60}$$

$\hat{f}_{\pi^t}^*$ is the reward function that minimizes the empirical ranking loss $\hat{L}_k$. Let $f_{\pi^t}^*$ be the solution to the true ranking loss $L_k$. As shown previously in Eq 30 and Eq 31, we can bound the expected values of these two quantities with high probability under agent or expert distribution.

We also have from concentration bound:

$$\mathbb{E}_{s,a\sim\rho_{\hat{M}_t}^{\pi^t}}\left[\hat{f}_{\pi^t}^*(s,a)\right] - \frac{\sum_{s,a\in\hat{\rho}^E}[\hat{f}_{\pi^t}^*(s,a)]}{|\hat{\rho}^E|} \leq \mathbb{E}_{s,a\sim\rho_{\hat{M}_t}^{\pi^t}}\left[\hat{f}_{\pi^t}^*(s,a)\right] - \mathbb{E}_{s,a\sim\rho_M^E}\left[\hat{f}_{\pi^t}^*(s,a)\right] + \epsilon_{stat} \tag{61}$$

Therefore, combining Eq 61 and Eq 59:

$$\mathbb{E}_{s,a\sim\rho_{\hat{M}_t}^{\pi^t}}\left[\hat{f}_{\pi^t}^*(s,a)\right] - \mathbb{E}_{s,a\sim\rho_M^E}\left[\hat{f}_{\pi^t}^*(s,a)\right] \geq \mathbb{E}_{s,a\sim\rho_{\hat{M}_t}^{\pi^E}}\left[\hat{f}_{\pi^E}^*(s,a)\right] - \mathbb{E}_{s,a\sim\rho_M^E}\left[\hat{f}_{\pi^E}^*(s,a)\right] - 2\epsilon_{stat} \tag{62}$$

The LHS of the Eq. 62 can be further upper bounded as follows:

$$\mathbb{E}_{s,a\sim\rho_{\hat{M}_t}^{\pi^t}}\left[\hat{f}_{\pi^t}^*(s,a)\right] - \mathbb{E}_{s,a\sim\rho_M^E}\left[\hat{f}_{\pi^t}^*(s,a)\right] \leq \mathbb{E}_{s,a\sim\rho_{\hat{M}_t}^{\pi^t}}[f_{\pi^t}^*(s,a)] - \mathbb{E}_{s,a\sim\rho_M^E}[f_{\pi^t}^*(s,a)] + 2R_{max}\sqrt{8\epsilon_r} \tag{63}$$

$$= \mathbb{E}_{s,a\sim\rho_{\hat{M}_t}^{\pi^t}}\left[\frac{k\rho_M^{\pi^E}(s,a)}{\rho_M^{\pi^E}(s,a) + \rho_{\hat{M}_t}^{\pi^t}(s,a)}\right]$$

$$- \mathbb{E}_{s,a\sim\rho_M^E}\left[\frac{k\rho_M^{\pi^E}(s,a)}{\rho_M^{\pi^E}(s,a) + \rho_{\hat{M}_t}^{\pi^t}(s,a)}\right] + 2R_{max}\sqrt{8\epsilon_r} \tag{64}$$

$$= k\mathbb{E}_{s,a\sim\rho_M^{\pi^E}}\left[\frac{\rho_{\hat{M}_t}^{\pi^t}(s,a) - \rho_M^{\pi^E}(s,a)}{\rho_{\hat{M}_t}^{\pi^t}(s,a) + \rho_M^{\pi^E}(s,a)}\right] + 2R_{max}\sqrt{8\epsilon_r} \tag{65}$$

$$= -kD_f(\rho_{\hat{M}_t}^{\pi^t}\|\rho_M^{\pi^E}) + 2R_{max}\sqrt{8\epsilon_r} \tag{66}$$

Similarly the RHS of Eq 62 can be further lower bounded as follows:

$$\mathbb{E}_{s,a \sim \rho^{\pi^E}_{\hat{M}_t}}\left[\hat{f}^*_{\pi^E}(s,a)\right] - \mathbb{E}_{s,a \sim \rho^E_M}\left[\hat{f}^*_{\pi}(s,a)\right] - 2\epsilon_{stat} \tag{67}$$

$$\geq \mathbb{E}_{s,a \sim \rho^{\pi^E}_{\hat{M}_t}}[f^*_{\pi^E}(s,a)] - \mathbb{E}_{s \sim \rho^E_M}[f^*_{\pi}(s,a)] - 2\epsilon_{stat} - 2R_{max}\sqrt{8\epsilon_r} \tag{68}$$

$$= k\mathbb{E}_{s,a \sim \rho^{\pi^E}_M}\left[\frac{\rho^{\pi^E}_{\hat{M}_t}(s,a) - \rho^{\pi^E}_M(s,a)}{\rho^{\pi^E}_{\hat{M}_t}(s,a) + \rho^{\pi^E}_M(s,a)}\right] - 2\epsilon_{stat} - 2R_{max}\sqrt{8\epsilon_r} \tag{69}$$

$$= -kD_f(\rho^E_{\hat{M}_t}\|\rho^E_M) - 2\epsilon_{stat} - 2R_{max}\sqrt{8\epsilon_r} \tag{70}$$

Plugging the relations obtained (Eq 70 and 66) back in Eq 62, we see that the $f$-divergence between the agent visitation in the learned MDP and the expert visitation in the ground truth MDP is bounded by the $f$-divergence of the expert policy's visitation on the learned vs. ground truth environment. We expect this term to be low if the dynamics are accurately learned at the transitions encountered in visitation of expert.

$$D_f(\rho^{\pi^t}_{\hat{M}_t}\|\rho^{\pi^E}_M) \leq D_f(\rho^{\pi^E}_{\hat{M}_t}\|\rho^{\pi^E}_M) + \frac{2\epsilon_{stat} + 4R_{max}\sqrt{8\epsilon_r}}{k} \tag{71}$$

We can use the total-variation lower bound for this $f$-divergence to later obtain a performance bound between the policy in learned MDP and expert in ground-truth MDP.

$$D_{TV}(\rho^{\pi^t}_{\hat{M}_t}\|\rho^{\pi^E}_M) \leq 2\sqrt{D_f(\rho^{\pi^t}_{\hat{M}_t}\|\rho^{\pi^E}_M)} \leq 2\sqrt{D_f(\rho^{\pi^E}_{\hat{M}_t}\|\rho^{\pi^E}_M) + \frac{2\epsilon_{stat} + 4R_{max}\sqrt{8\epsilon_r}}{k}} \tag{72}$$

Similar to Corollary A.1.1, we can further get a performance bound:

$$|V^{\pi^E}_M - V^{\pi^t}_{\hat{M}}| \leq \frac{2R_{max}}{1-\gamma}D_{TV}(\rho^{\pi^t}_{\hat{M}_t}\|\rho^{\pi^E}_M) \leq \frac{4R_{max}}{1-\gamma}\sqrt{D_f(\rho^{\pi^E}_{\hat{M}_t}\|\rho^{\pi^E}_M) + \frac{2\epsilon_{stat} + 4R_{max}\sqrt{8\epsilon_r}}{k}} \tag{73}$$

Let the local model error in the visitation of $\pi^t$ be bounded by $\epsilon^{\pi^t}_m$, i.e $\mathbb{E}_{s,a \sim \rho^{\pi^t}}\left[D_{TV}(P_M(.|s,a)\|P_{\hat{M}}(.|s,a))\right] \leq \epsilon^{\pi^t}_m$. Using simulation lemma for local models (Kearns & Singh, 1998; Kakade & Langford, 2002), we have:

$$|V^{\pi^t}_M - V^{\pi^t}_{\hat{M}}| \leq \frac{2\gamma\epsilon^{\pi^t}_m R_{max}}{(1-\gamma)^2} \tag{74}$$

We are interested in bounding the performance of the policy $\pi^t$ in ground-truth MDP rather than the learned MDP.

$$V^{\pi^E}_M - V^{\pi^t}_M \leq V^{\pi^E}_M - V^{\pi^t}_{\hat{M}} + \frac{4R_{max}}{1-\gamma}\sqrt{D_f(\rho^{\pi^E}_{\hat{M}_t}\|\rho^{\pi^E}_M) + \frac{2\epsilon_{stat} + 4R_{max}\sqrt{8\epsilon_r}}{k}} \tag{75}$$

$$\leq \frac{2\gamma\epsilon^{\pi^t}_m R_{max}}{(1-\gamma)^2} + \frac{4R_{max}}{1-\gamma}\sqrt{D_f(\rho^{\pi^E}_{\hat{M}_t}\|\rho^{\pi^E}_M) + \frac{2\epsilon_{stat} + 4R_{max}\sqrt{8\epsilon_r}}{k}} \tag{76}$$

The regret of an algorithm with ranking-loss depends on the accuracy of the approximate transition model at the visitation of the output policy $\pi_t$ and the expected accuracy of the approximate transition model at the transitions encountered in the visitation of expert. Using an exploratory policy optimization procedure, the regret grows sublinearly as shown in (Kidambi et al., 2021). Kidambi et al. (2021) uses an exploration bonus and shows that the RHS in the above regret simplifies to be information gain and for a number of MDP families the growth rate of information gain is mild. □

**Potential imitation suboptimality with additional rankings**

In this section, we consider how additional rankings can affect the intended performance gap as discussed in 4.2. Consider a tabular MDP setting in which we are given a set of rankings $\rho^\pi \preceq \rho^1 \preceq .. \preceq \rho^n \preceq \rho^E$. In such a case, we regress the state-action pairs from their respective visitations to $[0, k_1, k_2, ..k_n, k]$ where $0 < k_1 < k_2.. < k_n < k$. We will discuss in Appendix B.1.1 how this regression generalizes $L_k$ and make it computationally more efficient. For this regression, the optimal reward function that minimizes the ranking loss pointwise is given by:

$$R^*(s, a) = \frac{\sum_{i=1}^n k_i \rho^{\pi^i}(s, a) + \rho^E(s, a)}{\rho^\pi(s, a) + \sum_{i=1}^n \rho^{\pi^i}(s, a) + \rho^E(s, a)} \tag{77}$$

We consider a surrogate ranking loss with regression target $k_{eff}$ that achieves the same optimal reward when only $\rho \preceq \rho^E$ ranking is given. Therefore:

$$\frac{\sum_{i=1}^n k_i \rho^i(s, a) + k \rho^E(s, a)}{\rho^\pi(s, a) + \sum_{i=1}^n \rho^i(s, a) + \rho^E(s, a)} = \frac{k_{eff} \rho^E(s, a)}{\rho^E(s, a) + \rho^\pi(s, a)} \tag{78}$$

$k_{eff}$ can be upper bounded as follows:

$$k_{eff} = \frac{\rho^E(s, a) + \rho^\pi(s, a)}{\rho^E(s, a)} \frac{\sum_{i=1}^n k_i \rho^{\pi^i}(s, a) + k \rho^E(s, a)}{\rho^\pi(s, a) + \sum_{i=1}^n \rho^{\pi^i}(s, a) + k \rho^E(s, a)} \tag{79}$$

$$\leq \frac{\rho^E(s, a) + \rho^\pi(s, a)}{\rho^E(s, a)} \frac{\sum_{i=1}^n k_i \rho^{\pi^i}(s, a) + k \rho^E(s, a)}{\rho^\pi(s, a) + \rho^E(s, a)} \tag{80}$$

$$= k + \sum_{i=1}^n k_i \frac{\rho^{\pi^i}(s, a)}{\rho^E(s, a)} \tag{81}$$

$k_{eff}$ can be lower bounded by:

$$k_{eff} = \frac{\rho^E(s, a) + \rho^\pi(s, a)}{\rho^E(s, a)} \frac{\sum_{i=1}^n k_i \rho^{\pi^i}(s, a) + k \rho^E(s, a)}{\rho^\pi(s, a) + \sum_{i=1}^n \rho^{\pi^i}(s, a) + \rho^E(s, a)} \tag{82}$$

$$\geq \frac{\rho^E(s, a) + \rho^\pi(s, a)}{\rho^E(s, a)} \frac{k \rho^E(s, a)}{\rho^\pi(s, a) + \sum_{i=1}^n \rho^{\pi^i}(s, a) + \rho^E(s, a)} \tag{83}$$

$$= \frac{k}{1 + \frac{\sum_{i=1}^n \rho^{\pi^i}(s, a)}{\rho^\pi(s, a) + \rho^E(s, a)}} \tag{84}$$

Thus, $k_{eff}$ can increase or decrease compared to $k$ after augmenting the ranking dataset. We discuss the consequences of a decreased $k$ in Section 4.2.

# B  Algorithm Details

## B.1  Ranking Loss for the Reward Agent

Consider a dataset of behavior rankings $\mathcal{D} = \{(\rho_1^1 \preceq \rho_1^2), (\rho_2^1 \preceq \rho_2^2), ...(\rho_n^1 \preceq \rho_n^2)\}$, wherein for $\rho_j^i$ — $i$ denotes the comparison index within a pair of policies, $j$ denotes the pair number, and $\rho_1^1 \preceq \rho_1^2$ denotes that $\rho_1^2$ is preferable in comparison to $\rho_1^1$ and in turn implies that $\rho_1^2$ has a higher return. Each pair of behavior comparisons in the dataset are between the state-action or state visitations. We will restrict our attention to a specific instantiation of the ranking loss (a regression loss) that attempts to explain the rankings between each pair of policies present in the dataset by a performance gap of at least $k$, i.e. $\mathbb{E}_{\rho^1}[R(s, a)] \leq \mathbb{E}_{\rho^2}[R(s, a)] - k$. Formally, the ranking loss is defined as follows:

$$\min_R L_k(\mathcal{D}; R) = \min_R \mathbb{E}_{(\rho^1, \rho^2) \sim \mathcal{D}} \left[ \mathbb{E}_{s \sim \rho^1(s, a)} \left[ (R(s, a) - 0)^2 \right] + \mathbb{E}_{s \sim \rho^2(s, a)} \left[ (R(s, a) - k)^2 \right] \right] \tag{85}$$

When $k$ is set to 1 ($k = 1$), this loss function resembles the loss function used for SQIL (Reddy et al., 2019) if fixed rewards were used instead of learned. Thus, SQIL can be understood as a special case. We also note

that a similar ranking loss function has been previously used for training generative adversarial networks in LS-GAN (Mao et al., 2017).

Our work explores the setting of imitation learning given samples from state or state-action visitation $\rho^E$ of the expert $\pi^E$. We will use $\pi_m^{agent}$ to denote the $m^{th}$ update of the agent in Algorithm 1. The updated agent generates a new visitation in the environment which is stored in an empty dataset $\mathcal{D}_m^{online}$ given by $\mathcal{D}_m^{online} = \{\rho^{\pi_m^{agent}} \preceq \rho^{\pi^E}\}$

### B.1.1 Reward loss with automatically generated rankings (auto)

The ranking dataset $\mathcal{D}^p$ contains pairwise comparison between behaviors $\rho_i \preceq \rho_j$. First, we assume access to the trajectories that generate the behaviors, i.e $\rho^i = \{\tau_1^i, \tau_2^i...\tau_n^i\}$ and $\rho^j = \{\tau_1^j, \tau_2^j...\tau_m^j\}$ In this method we propose to automatically generate additional rankings using the following procedure: (a) Sample trajectory $\tau^i \sim \rho^i$ and $\tau^j \sim \rho^j$. Both trajectories are equal length because of our use of absorbing states (see Appendix C). (b) Generate an interpolation $\tau_{\lambda_p}^{ij}$ between trajectories depending on a parameter $\lambda_p$. A trajectory is a matrix of dimensions $H \times (|\mathcal{S}|+|\mathcal{A}|)$, where $H$ is the horizon length of all the trajectories.

$$\tau_{\lambda_p}^{ij} = \lambda_p \tau_i + (1 - \lambda_p)\tau_j \tag{86}$$

These intermediate interpolated trajectories lead to a ranking that matches the ranking under the expert reward function if the reward function is indeed linear in state features. We further note that $\tau$ can also be a trajectory of features rather than state-action pairs.

Next, we generate regression targets for the interpolated trajectories. For a trajectory $\tau_{\lambda_p}^{ij}$ the regression target is given by a vector $\lambda_p \mathbf{0} + (1 - \lambda_p)k\mathbf{1}$, where vectors $\mathbf{0}, \mathbf{1}$ are given by $[0,0,..0]$ and $[1,1,...,1]$ of length $H$ respectively. This procedure can be regarded as a form of mixup (Zhang et al., 2017) in trajectory space. The set of obtained $\tau_{\lambda_p}^{ij}$ after expending the sampling budget forms our behavior $\rho_{\lambda_p}^{ij}$.

**A generalized and computationally efficient interpolation strategy for `rank-game`**

Once we have generated $P$ interpolated rankings, we effectively have $O(P^2)$ rankings that we can use to augment our ranking dataset. Using them all naively would incur a high memory burden. Thus, we present another method for achieving the same objective of using automatically generated rankings in a more efficient and generalized way. For each pairwise ranking $\rho^i \preceq \rho^j$ in the dataset $\mathcal{D}^p$, we have the following new set of rankings $\rho^i \preceq \rho_{\lambda_1}^{ij} \preceq .. \preceq \rho_{\lambda_P}^{ij} \preceq \rho^j$. Using the $O(P^2)$ rankings in the ranking loss $L_k$, the ranking loss can be simplified to the following using basic algebraic manipulation:

$$(P+1)\mathbb{E}_{(s,a)\sim\rho^j}\left[(R(s,a)-k)^2\right] + (P)\mathbb{E}_{(s,a)\sim\rho_{\lambda_P}^{ij}}\left[(R(s,a)-k)^2\right] + .. + (1)\mathbb{E}_{(s,a)\sim\rho_{\lambda_1}^{ij}}\left[(R(s,a)-k)^2\right]$$
$$+(P+1)\mathbb{E}_{(s,a)\sim\rho^i}\left[(R(s,a)-0)^2\right] + (P)\mathbb{E}_{(s,a)\sim\rho_{\lambda_1}^{ij}}\left[(R(s,a)-0)^2\right] + .. + (1)\mathbb{E}_{(s,a)\sim\rho_{\lambda_P}^{ij}}\left[(R(s,a)-0)^2\right] \tag{87}$$

The reward function that minimizes the above loss pointwise is given by:

$$R^*(s,a) = \frac{k[(P+1)\rho^j + P\rho_{\lambda_P}^{ij} + (P-1)\rho_{\lambda_{P-1}}^{ij} + .. + \rho_{\lambda_1}^{ij}]}{(P+1)(\rho^j + \rho_{\lambda_P}^{ij} + .. + \rho_{\lambda_1}^{ij} + \rho^i)} \tag{88}$$

$$= \frac{k[\rho^j + \frac{P}{P+1}\rho_{\lambda_P}^{ij} + \frac{P-1}{P+1}\rho_{\lambda_{P-1}}^{ij} + .. + \frac{1}{P+1}\rho_{\lambda_1}^{ij}]}{(\rho^j + \rho_{\lambda_P}^{ij} + .. + \rho_{\lambda_1}^{ij} + \rho^i)} \tag{89}$$

We consider a modification to the ranking loss objective (Equation 85) that increases flexibility in regression targets for ranking as well as reducing the computational burden from dealing with $O(P^2)$ rankings pairs to $O(P)$. In this modification we regress the current agent, the expert, and each of the intermediate interpolants $(\rho^i, \rho_{\lambda_1}^{ij}, ..., \rho_{\lambda_P}^{ij}, \rho^E)$ to a fixed scalar return $(k_0, k_1, ..., k_{P+1})$ where $k_0 \le k_1 \le ... \le k_{P+1} = k$. The optimal reward function for this loss function is given by:

$$R^*(s,a) = \frac{k_{p+1}\rho^E(s,a) + k_p\rho_{\lambda_P}^{ij}(s,a) + k_{p-1}\rho_{\lambda_{P-1}}^{ij}(s,a) + .. + k_1\rho_{\lambda_1}^{ij}(s,a) + k_0\rho^\pi(s,a)}{(\rho^E(s,a) + \rho_{\lambda_P}^{ij}(s,a) + .. + \rho_{\lambda_1}^{ij}(s,a) + \rho^\pi)(s,a)} \tag{90}$$

This modified loss function generalizes Eq 88 and recovers it exactly when $[k_0, k_1.., k_{P+1}]$ is set to be $[0, k\frac{1}{P+1}, .., k\frac{P}{P+1}, k]$. We will call this reward loss function a *generalized ranking loss.*

**Shaping the ranking loss:** The generalized ranking loss contains a set of regression targets $(k_0, k_1, ..., k_{P+1})$ which needs to be decided apriori. We propose two strategies for deciding these regression targets. We consider two families of parameterized mappings: (1) linear in $\alpha$ ($k_\alpha = \alpha * k$) and (2) rate of increase in return exponential in $\alpha$ ($\frac{dk_\alpha}{d\alpha} \propto e^{\beta\alpha}$), where $\beta$ is the temperature parameter and denote this family by exp-$\beta$. We also set $k_{\alpha=0} = 0$ (in agent's visitation) and $k_{\alpha=1} = k$ (in expert's visitation) under the reward function that is bounded in $[0, R_{max}]$. The shaped ranking regression loss, denoted by $SL_k(\mathcal{D}; R)$, that induces a performance gap between $p + 2$ consecutive rankings ($\rho^i = \rho^{ij}_{\lambda_0}, \rho^{ij}_{\lambda_1}, ..., \rho^{ij}_{\lambda_P}, \rho^j = \rho^{ij}_{\lambda_{P+1}}$) is given by:

$$SL_k(\mathcal{D}; R) = \frac{1}{p+2} \sum_{i=0}^{p+1} \mathbb{E}_{s \sim \rho^{ij}_{\lambda_i}(s,a)} \left[ (R(s,a) - k_i)^2 \right] \tag{91}$$

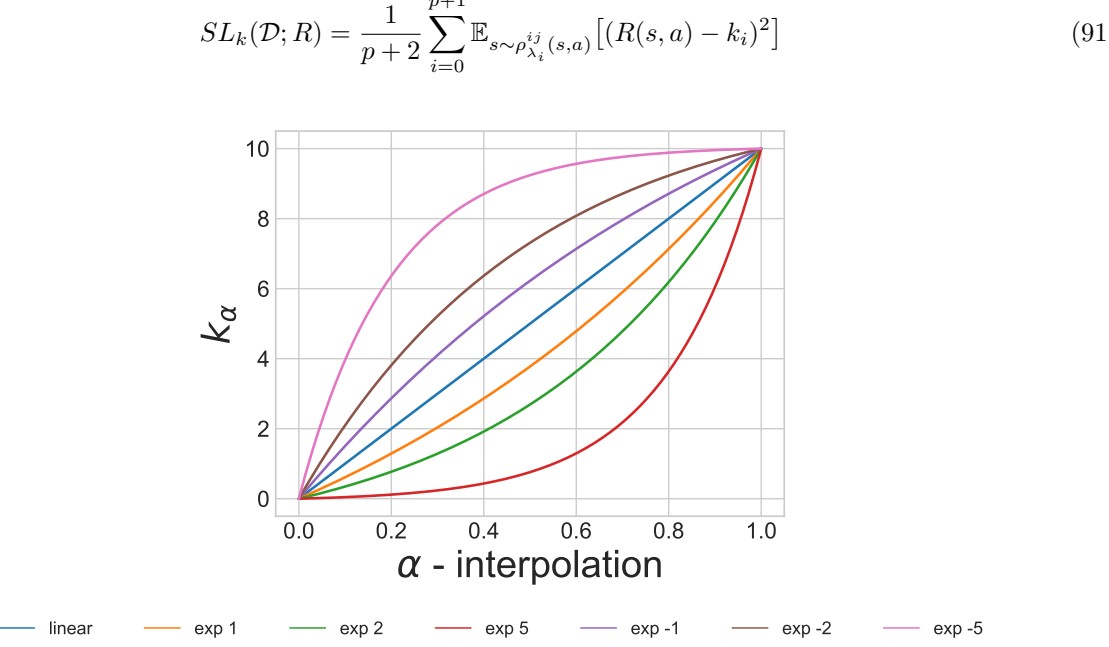

Figure 7: This figure shows the assignment of value $k_\alpha$ (intended return value) corresponding to values of $\alpha$ (degree of time-conditional interpolation between the visitation distribution of the agent and the expert). When the rate of increase is exponential with positive slope, we have a higher performance gap over comparisons closer to the expert and when the rate of increase is negative, the performance gap is higher for comparisons closer to the agent.

Figure 7 above shows the flexibility in reward shaping afforded by the two families of parameterized functions. The temperature parameter $\beta > 0$ encourages the initial preferences to have a smaller performance gap than the latter preferences. Conversely, $\beta < 0$ encourages the initial preferences to have a larger performance gap compared to the latter preferences. We ablate these choices of parameteric functions in Appendix D.5.

### B.1.2 Reward loss with offline annotated rankings (pref)

Automatically generated rankings are generated without any additional supervision and can be understood as a form of data augmentation. By contrast, with offline annotated rankings, we are given a fixed dataset of comparisons which is a form of additional supervision for the reward function. Automatically generated rankings can only help by making the reward landscape easier to optimize, but offline rankings can help reduce the exploration burden by informing the agent about counterfactuals that it had no information about. This can, for instance, help the agent avoid unnecessary exploration by providing a dense improvement signal. The offline rankings are either provided by a human or extracted from a set of trajectories for which ground truth reward is known. In our work, we extract offline preferences by uniformly sampling $p$ trajectories from an offline dataset obtained from a training run of an RL method (SAC) (Haarnoja et al., 2018) with ground truth reward.

For imitation learning with offline annotated rankings, at every iteration $m$ of Algorithm 1 we have a new dataset of rankings given by $\mathcal{D}_m^{online} = \{\rho_m^{agent} \preceq \rho^E\}$ along with a fixed offline dataset containing rankings of the form $(\mathcal{D}^{offline} = \{\rho^1 \preceq \rho^2 ... \preceq \rho^p\})$. We always ground the offline preferences by expert's visitation in our experiments, i.e $\rho^p \preceq \rho^E$. We incorporate the offline rankings as a soft constraint in reward learning by combining the ranking loss $L_k$ between the policy agent and the expert, with a ranking loss $L_k$ or a shaped ranking loss $SL_k$ (Equation 91) over offline trajectories:

$$L_k^{offline}(\mathcal{D}^{online}, \mathcal{D}^{offline}; R) = \alpha L_k(\mathcal{D}^{online}; R) + (1 - \alpha) * L_k(\mathcal{D}^{offline}; R) \tag{92}$$

Here, instead of the consecutive rankings being interpolants, they are offline rankings. The videos attached in the supplementary show the benefit of using preferences in imitation learning. The policy learned without preferences in the pen environment drops the pen frequently and in the door environment is unable to successfully open the door.

## B.2 Stackelberg Game Instantiation

A Stackelberg game view of optimizing the two-player game with a dataset of behavior rankings leads to two methods: PAL (Policy as Leader) and RAL (Reward as Leader) (refer Section 4.3). PAL uses a fast reward update step and we simulate this step by training the reward function until convergence (using a validation set) on the dataset of rankings. We simulate a slow update step of the policy by using a few iterations of the SAC (Haarnoja et al., 2018) update for the policy. RAL uses a slow reward update which we approximate by dataset aggregation — aggregating all the datasets of rankings generated by the agent in each previous iteration enforces the reward function to update slowly. A fast policy update is simulated by using more iterations of SAC. Since SAC does not perform well with a high update to environment step ratio, more iterations of SAC would imply more environment steps under a fixed reward function. This was observed to lead to reduced learning efficiency, and an intermediate value of SAC updates was observed to perform best (Table 5).

### B.2.1 Policy as Leader

Algorithm 2 presents psuedocode for a practical instantiation of the PAL methods - RANK-PAL (vanilla), RANK-PAL (auto) and RANK-PAL (pref) that we use in our work. Recall that (vanilla) variant uses no additional rankings, whereas (auto) uses automatically generated rankings and (pref) uses offline annotated ranking.

### B.2.2 Reward as Leader

Algorithm 3 presents psuedocode for a practical instantiation of the RAL methods - RANK-RAL (vanilla), RANK-RAL (auto).

## C Implementation and Experiment Details

**Environments:** Figure 8 shows some of the environments we use in this work. For benchmarking we use 6 MuJoCo (licensed under CC BY 4.0) locomotion environments. We also test our method on manipulation environments - Door opening environment from Robosuite (Zhu et al., 2020a) (licensed under MIT License) and the Pen-v0 environment from mjrl (Rajeswaran et al., 2017) (licensed under Apache License 2.0).

**Expert data**: For all environments, we obtain expert data by a policy trained until convergence using SAC (Haarnoja et al., 2018) with ground truth rewards.

**Baselines:** We compare our proposed methods against 6 representative LfO approaches that cover a spectrum of on-policy and off-policy, model-free methods from prior work: GAIfO (Torabi et al., 2018b; Ho & Ermon, 2016), DACfO (Kostrikov et al., 2018), BCO (Torabi et al., 2018a), $f$-IRL (Ni et al., 2020), OPOLO (Zhu et al., 2020b) and IQ-Learn Garg et al. (2021). GAIfO (Torabi et al., 2018b) is a modification of the adversarial GAIL method (Ho & Ermon, 2016), in which the discriminator is trained to distinguish between

---

**Algorithm 2** Policy As Leader (PAL) practical instantiation

---

1: **Initialize:** Policy network $\pi_\theta$, reward network $R_\phi$, replay buffer $\mathcal{R}$
2: **Hyperparameters:** *Common*: Policy update steps $n_{pol}$, Reward update steps $n_{rew}$, Performance gap $k$, empty ranking dataset $\mathcal{D}^{online}$, *RANK-PAL (auto)*: number of interpolations $P$, *RANK-PAL(pref)*: Offline annotated rankings $\mathcal{D}^{offline}$.
3: **for** $m = 0, 1, 2, \ldots$ **do**
4:     Collect transitions in the environment and add to replay buffer $\mathcal{R}$. Run policy update step: $\pi_\theta^m = $ Soft Actor-Critic($R_\phi^{m-1}; \pi_\theta^{m-1}$) with transitions relabelled with reward obtained from $R_\phi^{m-1}$. `// call` $n_{pol}$ `times`
5:     Add absorbing state/state-actions to all early-terminated trajectories collected in the current $n_{pol}$ policy update steps to make them full horizon and collect in $\mathcal{D}_m^{online}$. $\mathcal{D}^{online} = \mathcal{D}_m^{online}$ (discard old data).
6:     (for RANK-PAL(auto)) Generate interpolations for rankings in the dataset $\mathcal{D}^{online}$ and collect in $\mathcal{D}_{auto}^{online}$
7:     Reward Update step: `// call` $n_{rew}$ `times`

$$R_\phi^m = \begin{cases} \min L_k(\mathcal{D}^{online}; R_\phi^{m-1}), & \text{RANK-PAL (vanilla) (Equation 85)} \\ \min SL_k(\mathcal{D}_{auto}^{online}; R_\phi^{m-1}), & \text{RANK-PAL (auto) (Equation 91)} \\ \min L_k^{offline}(\mathcal{D}^{online}, \mathcal{D}^{offline}; R), & \text{RANK-PAL (pref) (Equation 92)} \end{cases}$$

8: **end for**

---

**Algorithm 3** Reward As Leader (RAL) practical instantiation

---

1: **Initialize:** Policy network $\pi_\theta$, reward network $R_\phi$, replay buffer $\mathcal{R}$, trajectory buffer $D$
2: **Hyperparameters:** *Common*: Policy update steps $n_{pol}$, Reward update steps $n_{rew}$, Performance gap $k$, empty ranking dataset $\mathcal{D}^{online}$, *RANK-PAL (auto)*: number of interpolations $P$, *RANK-PAL (pref)*: Offline annotated rankings $\mathcal{D}^{offline}$.
3: **for** $m = 0, 1, 2, \ldots$ **do**
4:     Collect transitions in the environment and add to replay buffer $\mathcal{R}$. Run policy update step: $\pi_\theta^m = $ Soft Actor-Critic($R_\phi^{m-1}; \pi_\theta^{m-1}$) with transitions relabelled with reward obtained from $R_\phi^{m-1}$. `// call` $n_{pol}$ `times`
5:     Add absorbing state/state-actions to all early-terminated trajectories collected in the current $n_{pol}$ policy update steps to make them full horizon and collect in $D_m^{online}$. Aggregate data in $\mathcal{D}^{online} = \mathcal{D}_m^{online} \cup \mathcal{D}^{online}$.
6:     (for RANK-RAL(auto)) Generate interpolations for rankings in the dataset $\mathcal{D}^{online}$ and collect in $\mathcal{D}_{auto}^{online}$
7:     Reward Update step: `// call` $n_{rew}$ `times`

$$R_\phi^m = \begin{cases} \min L_k(\mathcal{D}^{online}; R_\phi^{m-1}), & \text{RANK-RAL (vanilla) (Equation 85)} \\ \min SL_k(\mathcal{D}_{auto}^{online}, R_\phi^{m-1}), & \text{RANK-RAL(auto) (Equation 91)} \end{cases}$$

8: **end for**

---

state-distributions rather than state-action distributions. DAC-fO (Kostrikov et al., 2018) is an off-policy modification of GAIfO (Torabi et al., 2018b), in which the discriminator distinguishes the expert states with respect to the entire replay buffer of the agent's previously visited states, with additional implementation details such as added absorbing states to early-terminated trajectories. BCO (Torabi et al., 2018a) learns an inverse dynamics model, iteratively using the state-action-next state visitation in the environment and using it to predict the actions that generate the expert state trajectory. OPOLO (Zhu et al., 2020b) is a recent method which presents a principled off-policy approach for imitation learning by minimizing an upper-bound of the state marginal matching objective. IQ-Learn (Garg et al., 2021) proposes to make

| Env | Swimmer | Hopper | HalfCheetah | Walker | Ant | Humanoid |
|---|---|---|---|---|---|---|
| BCO | 102.76±0.90 | 20.10±2.15 | 5.12±3.82 | 4.00±1.25 | 12.80±1.26 | 3.90±1.24 |
| GaiFO | 99.04±1.61 | 81.13± 9.99 | 13.54±7.24 | 83.83±2.55 | 20.10±24.41 | 3.93±1.81 |
| DACfO | 95.09±6.14 | 94.73±3.63 | 85.03±5.09 | 54.70±44.64 | 86.45±1.67 | 19.31±32.19 |
| $f$ -IRL | 103.89±2.37 | 97.45± 0.61 | 96.06±4.63 | **101.16±1.25** | 71.18±19.80 | 77.93±6.372 |
| OPOLO | 98.64±0.14 | 89.56±5.46 | 88.92±3.20 | 79.19±24.35 | 93.37± 3.78 | 24.87±17.04 |
| IMIT-PAL (ours) | **105.93±3.12** | 86.47± 7.66 | 90.65±15.17 | 75.60±1.90 | 82.40±9.05 | 94.49±3.21 |
| IMIT-RAL (ours) | 100.35±3.6 | 92.34±8.63 | 96.80±2.45 | 94.41±2.94 | 78.06±4.24 | 91.27±9.33 |
| RANK-PAL (ours) | 98.83±0.09 | 87.14± 16.14 | 94.05±3.59 | 93.88±0.72 | **98.93±1.83** | **96.84±3.28** |
| RANK-RAL (ours) | 99.31±1.50 | **99.34±0.20** | **101.14±7.45** | 93.24±1.25 | 93.21±2.98 | 94.45±4.13 |
| Expert | 100.00± 0 | 100.00± 0 | 100.00± 0 | 100.00± 0 | 100.00± 0 | 100.00± 0 |
| $(|\mathcal{S}|, |\mathcal{A}|)$ | $(8, 2)$ | $(11, 3)$ | $(17, 6)$ | $(17, 6)$ | $(111, 8)$ | $(376, 17)$ |

Table 3: Asymptotic normalized performance of LfO methods at 2 million timesteps on MuJoCo locomotion tasks. The results in this Table also include evaluations for the IMIT-{PAL, RAL} methods.

| Env | Swimmer | Hopper | HalfCheetah | Walker | Ant | Humanoid |
|---|---|---|---|---|---|---|
| BCO | 210.22±3.43 | 721.92±89.89 | 410.83±238.02 | 224.58±71.42 | 704.88±13.49 | 324.94±44.39 |
| GAIfO | 202.66±4.87 | 2871.47±365.73 | 1532.57±693.72 | 4666.31±143.75 | 1141.66±1400.11 | 326.69±13.26 |
| DACfO | 194.65±14.08 | 3350.55±141.69 | 11057.54±407.26 | 3045.21±2485.33 | 5112.15±38.01 | 1165.40±1867.61 |
| $f$ -IRL | 212.50±6.43 | 3446.33±35.66 | 12527.24±344.95 | **5630.32±71.35** | 4200.48±1124.17 | 4362.46±459.72 |
| OPOLO | 210.84±1.31 | 3168.35±206.26 | 11576.12±155.09 | 4407.70±1356.39 | 5529.44±164.94 | 1468.90± 1041.853 |
| IMIT-PAL (ours) | **216.64±7.95** | 3059.43±283.85 | 11806.47± 1750.24 | 4208.17±107.41 | 4872.39±480.23 | 5265.60±287.44 |
| IMIT-RAL (ours) | 205.33±8.92 | 3266.28±318.03 | 12626.18±54.71 | 5254.54±165.19 | 4612.8±192.06 | 5089.88±621.07 |
| RANK-PAL (ours) | 202.24±1.80 | 3082.98±582.59 | 12259.06± 206.82 | 5225.49±42.02 | **5862.42±47.68** | **5393.45±291.16** |
| RANK-RAL (ours) | 203.20±4.65 | **3512.67±21.09** | **13204.49±721.77** | 5189.51±71.27 | 5520.14±116.77 | 5262.96±337.44 |
| Expert | 204.6 ± 0 | 3535.88 ± 0 | 13051.46 ± 0 | 5456.91 ± 0 | 5926.17 ± 0 | 5565.53 ± 0 |
| $(|\mathcal{S}|, |\mathcal{A}|)$ | $(8, 2)$ | $(11, 3)$ | $(17, 6)$ | $(17, 6)$ | $(111, 8)$ | $(376, 17)$ |

Table 4: Asymptotic performance of LfO methods at 2 million timesteps on MuJoCo locomotion tasks. The results in this Table also include evaluations for the IMIT-{PAL, RAL} methods.

imitation learning non-adversarial by directly optimizing the Q-function and removing the need to learn a reward as a subproblem. All the approaches only have access to expert state-trajectories.

We use the author's open-source implementations of baselines OPOLO, DACfO, GAIfO, BCO available at `https://github.com/illidanlab/opolo-code`. We use the author-provided hyperparameters (similar to those used in (Zhu et al., 2020b)) for all MuJoCo locomotion environments. For $f$-IRL, we use the author implementation available at `https://github.com/twni2016/f-IRL` and use the author provided hyperparameters. IQ-Learn was tested on our expert dataset by following authors implementation found here: `https://github.com/Div99/IQ-Learn`. We tested two IQ-Learn loss variants: 'v0' and 'value' as found in their hyperparameter configurations and took the best out of the two runs.

**Policy Optimization:** We implement RANK-PAL and RANK-RAL with policy learning using SAC (Haarnoja et al., 2018). We build upon the SAC code (Achiam, 2018) (https://github.com/openai/spinningup) without changing any hyperparameters.

**Reward Learning:** For reward learning, we use an MLP parameterized by two hidden layers of 64 dimensions each. Furthermore, we clip the outputs of the reward network between $[-10, 10]$ range to keep the range of rewards bounded while also adding an L2 regularization of 0.01. We add absorbing states to early terminated agent trajectories following Kostrikov et al. (2019). For training the ranking loss until convergence in both

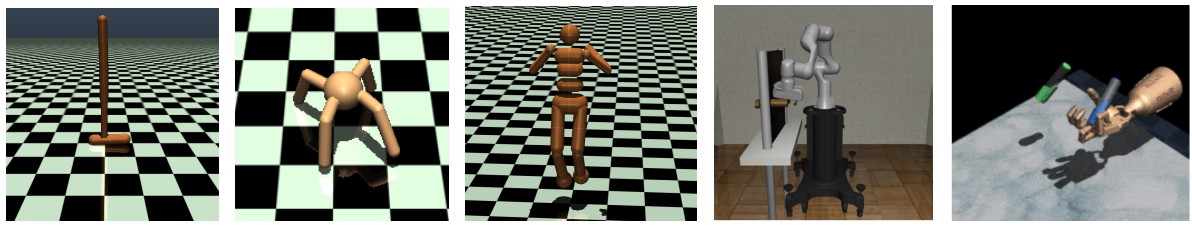

Figure 8: We evaluate `rank-game` over environments including Hopper-v2, Ant-v2, Humanoid-v2, Door, and Pen-v0.

update strategies (PAL and RAL), we used evaluation on a holdout set that is 0.1 the total dataset size as a proxy for convergence.

**Data sharing between players**: We rely on data sharing between players to utilize the same collected transitions for both players' gradient updates. The reward learning objective in RANK-PAL and RANK-RAL requires rolling out the current policy. This makes using an off-policy routine for training the policy player quite inefficient, since off-policy model-free algorithms update a policy frequently even when executing a trajectory. To remedy this, we reuse the data collected with a mixture of policies obtained during the previous off-policy policy learning step for training the reward player. This allows us to reuse the same data for policy learning as well as reward learning at each iteration.

**Ranking loss for reward shaping via offline annotated rankings**: In practice for the (pref) setting (Section 4.2), to increase supervision and prevent overfitting, we augment the offline dataset by regressing the snippets (length $l$) of each offline trajectory $\tau^i$ for behavior $\rho^i$ to $k*l$, in addition to regressing the rewards for each state to $k$. The snippets are generated as contiguous subsequence from the trajectory, similar to Brown et al. (2019).

### C.1  Hyperparameters

Hyperparameters for RANK-{PAL,RAL} (vanilla,auto and pref) methods are shown in Table 5. For RANK-PAL, we found the following hyperparameters to give best results: $n_{pol} = H$ and $n_{rew} = $ ('validation' or $H/b$), where H is the environment horizon (usually set to 1000 for MuJoCo locomotion tasks) and b is the batch size used for the reward update. For RANK-RAL, we found $n_{pol} = H$ and $n_{rew} = $ ('validation' or $|D|/b$), where $|D|$ indicates the cumulative size of the ranking dataset. We found that scaling reward updates proportionally to the size of the dataset also performs well and is a computationally effective alternative to training the reward until convergence (see Section D.7).

| Hyperparameter | Value |
|---|---|
| Policy updates $n_{pol}$ | H |
| Reward batch size($b$) | 1024 |
| Reward gradient updates $n_{rew}$ | val or $|D|/1024$ |
| Reward learning rate | 1e-3 |
| Reward clamp range | [-10,10] |
| Reward l2 weight decay | 0.0001 |
| Number of interpolations [auto] | 5 |
| Reward shaping parameterization [auto] | exp-[-1] |
| Offline rankings loss weight ($\lambda$) [pref] | 0.3 |
| Snippet length $l$ [pref] | 10 |

Table 5: Common hyperparameters for the RANK-GAME algorithms. Square brackets in the left column indicate which hyperparameters that are specific to 'auto' and 'pref' methods.

# D   Additional Experiments

## D.1   Complete evaluation of LfO with `rank-game(auto)`

Figure 9 shows a comparison of RANK-PAL(auto) and RANK-RAL(auto) for the LfO setting on the Mujoco benchmark tasks: `Swimmer-v2`, `Hopper-v2`, `HalfCheetah-v2`, `Walker2d-v2`, `Ant-v2` and `Humanoid-v2`. This section provides complete results for Section 5.1 in the main paper.

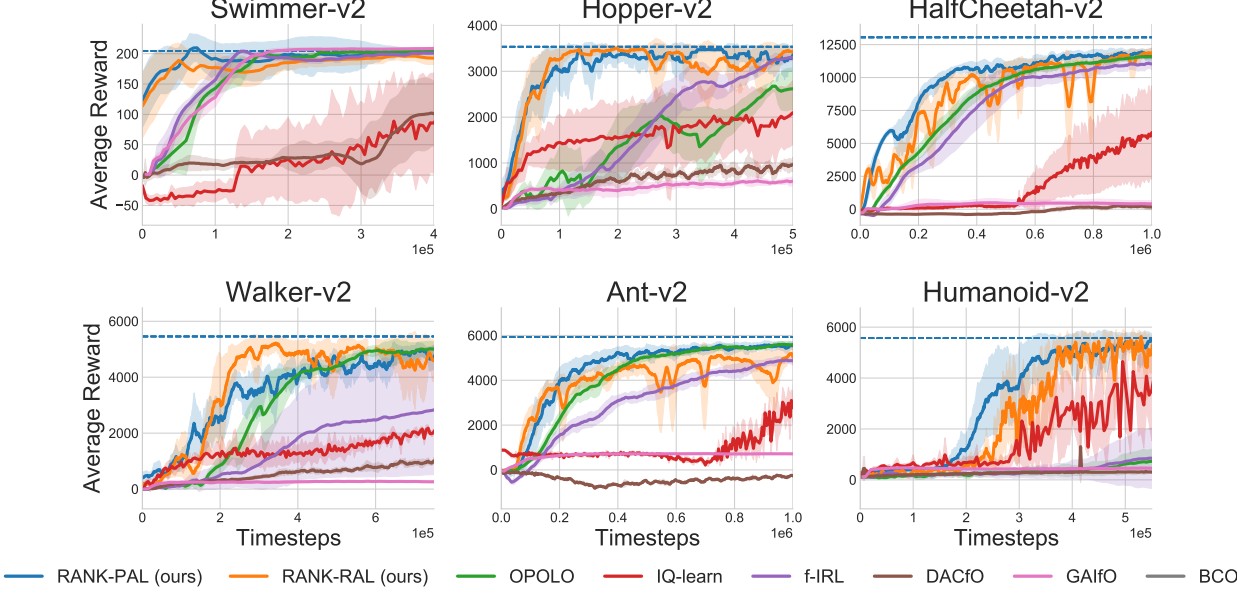

Figure 9: Comparison of performance on OpenAI gym benchmark tasks. The shaded region represents standard deviation across 5 random runs. RANK-PAL and RANK-RAL substantially outperform the baselines in sample efficiency. Dotted blue line shows the expert's performance.

## D.2   Evaluation of LfD with `rank-game(auto)`

`rank-game` is a general framework for both LfD(with expert states and actions) and LfO (with only expert states/observations). We compare performance of `rank-game` compared to LfD baselines: IQ-Learn (Garg et al., 2021), DAC (Kostrikov et al., 2018) and BC (Pomerleau, 1991).

In figure 10, we observe that `rank-game` is among the most sample efficient methods for learning from demonstrations. IQlearn shows poor learning performance on some tasks which we suspect is due to the low number of expert trajectories we use in our experiments compared to the original work. DAC was tuned using the guidelines from Orsini et al. (2021) to ensure fair comparison.

## D.3   Utility of automatically generated rankings in `rank-game(auto)`

We investigate the question of how much the automatically generated rankings actually help in this experiment. To do that, we keep all the hyperparameters same and compare RANK-GAME (vanilla) with RANK-GAME (auto). RANK-GAME (vanilla) uses no additional ranking information and $L_k$ is used as the reward loss.

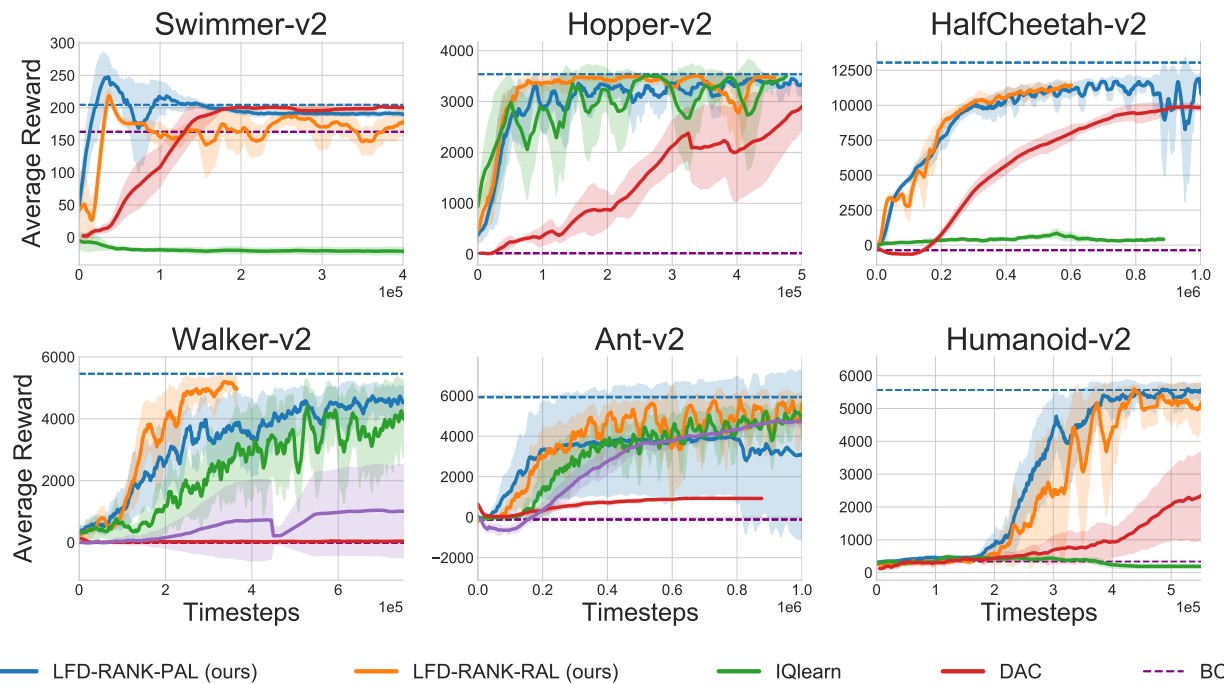

Figure 10: Comparison of `rank-game` methods with baselines in the LfD setting (expert actions are available). RANK-{PAL,RAL} are competitive to state of the art methods.

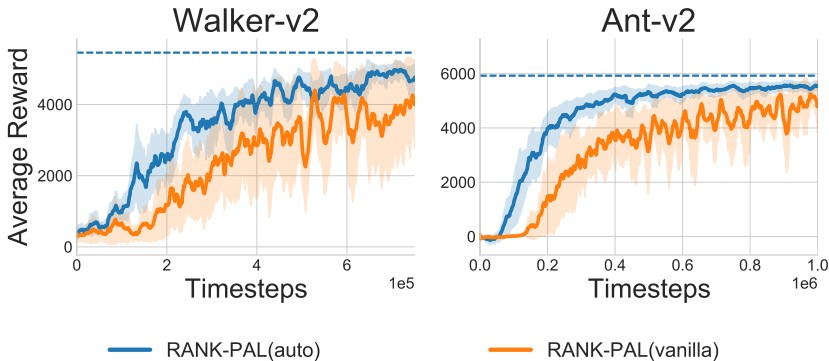

Figure 11: RANK-PAL(vanilla) has high variance learning curves with lower sample efficiency compared to RANK-PAL(auto).

Figure 11 shows that in RANK-PAL (auto) has lower variance throughout training (more stable) and is more sample efficient compared to RANK-PAL(vanilla).

## D.4 Comparison of `imit-game` and `rank-game` methods

Imitation learning algorithms, particularly adversarial methods, have a number of implementation components that can affect learning performance. In this experiment, we aim to further reduce any implementation/hyperparameter gap between adversarial imitation learning (AIL) methods that are based on the *supremum*-loss (described in section 3) function and `rank-game` to bring out the obtained algorithmic improvements. To achieve this, we swap out the ranking loss $L_k$ based on regression with a *supremum*-loss and call this method IMIT-{PAL,RAL}. This results in all the other hyperparameters such as batch size,

reward clipping, policy and reward learning iterations, and optimizer iterations to be held constant across experiments.

We present a comparison of RANK-{PAL, RAL} and IMIT-{PAL, RAL} in terms of asymptotic performance in Table 3 and their sample efficiency in Figure 12. Note that Table 3 shows normalized returns that are mean-shifted and scaled between [0-100] using the performance of a uniform random policy and the expert policy. The expert returns are given in Table 4 and we use the following performance values from random policies for normalization: { Hopper= 13.828, HalfCheetah= −271.93, Walker= 1.53, Ant= −62.01, Humanoid= 112.19}. Table 4 shows unnormalized asymptotic performance of the different methods.

In terms of sample efficiency, we notice IMIT-{PAL, RAL} methods compare favorably to other regularized *supremum*-loss counterparts like GAIL and DAC but are outperformed by RANK-{PAL, RAL} (auto) methods. We hypothesize that better learning efficiency in $L_k$ compared to *supremum*-loss is due to regression to fixed targets being a simpler optimization than maximizing the expected performance gap under two distributions.

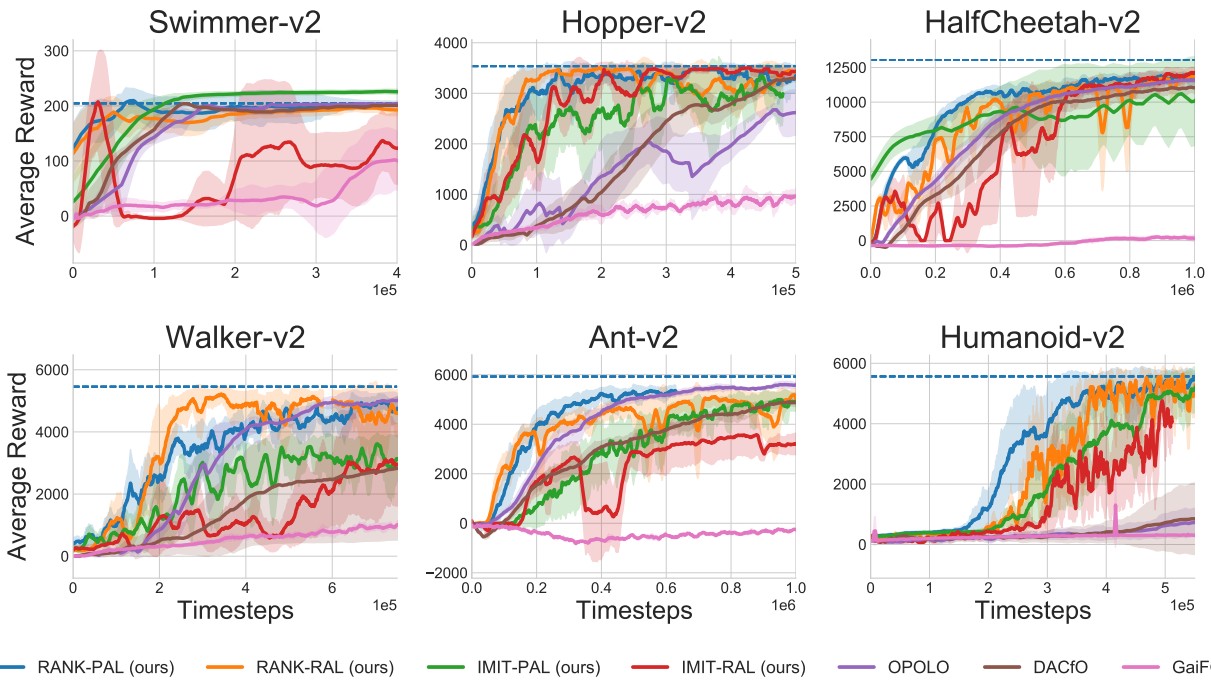

Figure 12: Comparison of performance on OpenAI gym benchmark tasks. Specifically, we seek to compare RANK-{PAL, RAL} methods to IMIT-{PAL, RAL} methods and IMIT-{PAL, RAL} methods to their non-Stackelberg counterparts GAIfO and DACfO. The shaded region represents standard deviation across 5 random runs. RANK-PAL and RANK-RAL substantially outperform the baselines in sample efficiency and IMIT-{PAL, RAL} is competitive to the strongest prior baseline OPOLO.

### D.5 Effect of parameterized reward shaping in `rank-game` (auto)

We experiment with different ways of shaping the regression targets (Appendix B) for automatically generated interpolations in RANK-GAME (auto) in Figure 13. In the two left-most plots for RANK-PAL (auto), we see that reward shaping instantiations (exponential with negative temperature) which learns a higher performance gap for pairs of interpolants closer to the agent lead to higher sample efficiency. We note that decreasing the temperature too much leads to a fall in sample efficiency. The same behavior is observed in RANK-RAL (two right-most plots) methods but we find them to be more robust to parameterized shaping than PAL methods. We use the following interpolation scheme: exponential with temperature=−1 for our experiments in the main paper.

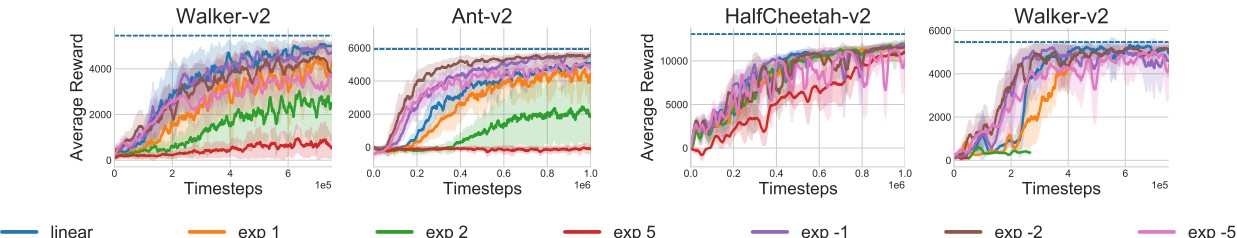

Figure 13: The two left-most plots show the effect of reward shaping in RANK-PAL (auto) methods using linear and exponential shaping functions. The two right-most plots show the same effect of reward shaping in RANK-RAL (auto) methods. Reward shaping instantiations which induce a higher performance gap between pairs of interpolants closer to the agent perform better and RAL is more robust to reward shaping variants than PAL.

## D.6   On the rank preserving nature of $SL_k$

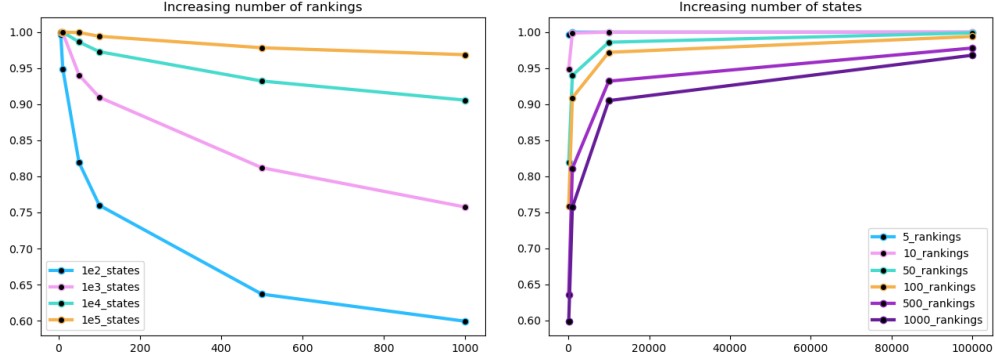

Figure 14: Increasing the state size of the domain increases the rank consistency afforded by $SL_k$ and increasing the number of rankings decreases the rank consistency.

The ranking loss $SL_k$ (Appendix B, Eq 91) regresses the $\rho^i$, $\rho^j$ and each of the intermediate interpolants $(\rho^i = \rho^{ij}_{\lambda_0}, \rho^{ij}_{\lambda_1}, ..., \rho^{ij}_{\lambda_P}, \rho^j = \rho^{ij}_{\lambda_{P+1}})$ to fixed scalar returns $(k_0, k_1, ..., k_{P+1})$ where $k_0 \leq k_1 \leq ... \leq k_{p+1} = k$. The ranking loss $SL_k$ is given by:

$$SL_k(\mathcal{D}; R) = \frac{1}{p+2} \sum_{i=0}^{P+1} \mathbb{E}_{s \sim \rho^{ij}_{\lambda_i}(s,a)} \big[ (R(s,a) - k_i)^2 \big] \tag{93}$$

$SL_k$ provides a dense reward assignment for the reward agent but does not guarantee that minimizing $SL_k$ would lead to the performance ordering between rankings, i.e $\mathbb{E}_{\rho^1}[f(s)] < \mathbb{E}_{\rho^2}[f(s)] < \mathbb{E}_{\rho^3}[f(s)] < .. < \mathbb{E}_{\rho^{P+1}}[f(s)]$. An ideal loss function for this task regresses the expected return under each behavior to scalar values indicative of ranking, but needs to solve a complex credit assignment problem. Formally, we can write the ideal loss function for reward agent as follows

$$SL_k^{ideal}(\mathcal{D}; R) = \frac{1}{p+2} \sum_{i=0}^{P+1} \big[ \mathbb{E}_{s \sim \rho^{ij}_{\lambda_i}(s,a)}[R(s,a)] - k_i \big]^2 \tag{94}$$

We note that the $SL_k$ upper bounds $SL_k^{ideal}$ using Jensen's inequality and thus is a reasonable target for optimization. In this section we wish to further understand if $SL_k$ has a rank-preserving policy. $SL_k$ is a family of loss function for ranking that assigns a scalar reward value for each states of a particular state visitation corresponding to its ranking. Ideally, given a ranking between behaviors $\rho^0 \preceq \rho^1 \preceq \rho^2 ... \preceq \rho^{P+1}$ we aim to learn a reward function $f$ that satisfies $\mathbb{E}_{\rho^0}[f(s)] < \mathbb{E}_{\rho^1}[f(s)] < \mathbb{E}_{\rho^2}[f(s)] < .. < \mathbb{E}_{\rho^{P+1}}[f(s)]$. We empirically test the ability of the ranking loss function $SL_k$ to facilitate the desired behavior in performance ranking. We consider a finite state space $\mathcal{S}$ and number of rankings $P$. We uniformly sample $P + 1$ possible state

visitations and the intermediate regression targets $\{k_i\}_{i=1}^{n}$ $s.t$ $k_i \leq k_{i+1}$. To evaluate the rank-preserving ability of our proposed loss function we study the fraction of comparisons the optimization solution that minimizes $SL_k$ is able to get correct. Note that $P+1$ sequential ranking induces $P(P+1)/2$ comparisons.

Figure 14 shows that with large state spaces $SL_k$ is almost rank preserving and the rank preserving ability degrades with increasing number of rankings to be satisfied.

## D.7 Stackelberg game design

We consider the sensitivity of the two-player game with respect to policy update iterations and reward update iterations. Our results (Figure 15) draw analogous conclusions to Rajeswaran et al. (2020) where we find that using a validation loss for training reward function on on-policy and aggregate dataset in PAL and RAL respectively works best. Despite its good performance, validation loss based training can be wall-clock inefficient. We found a substitute method to perform similarly while giving improvements in wall-clock time - make number of iterations of reward learning scale proportionally to the dataset set size. A proportionality constant of (1/batch-size) worked as well as validation loss in practice. Contrary to Rajeswaran et al. (2020) where the policy is updated by obtaining policy visitation samples from the learned model, our ability to increase the policy update is hindered due to unavailability of a learned model and requires costly real-environment interactions. We tune the policy iteration parameter (Figure 16) and observe the increasing the number of policy updates can hinder learning performance.

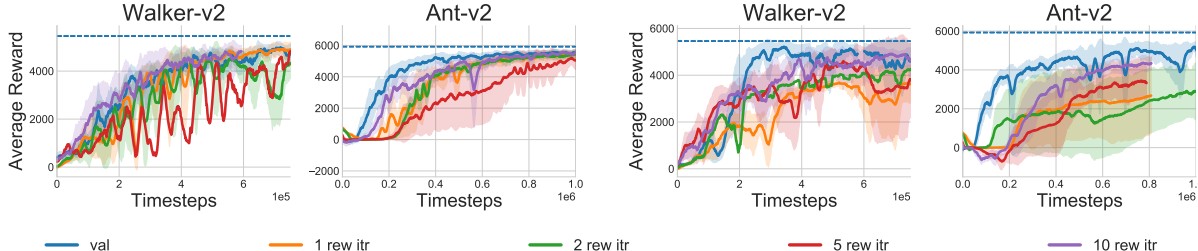

Figure 15: The left two plots use PAL strategy and the right two plots use RAL strategy. Reward learning using a validation loss on a holdout set leads to improved learning performance compared to hand designed reward learning iterations.

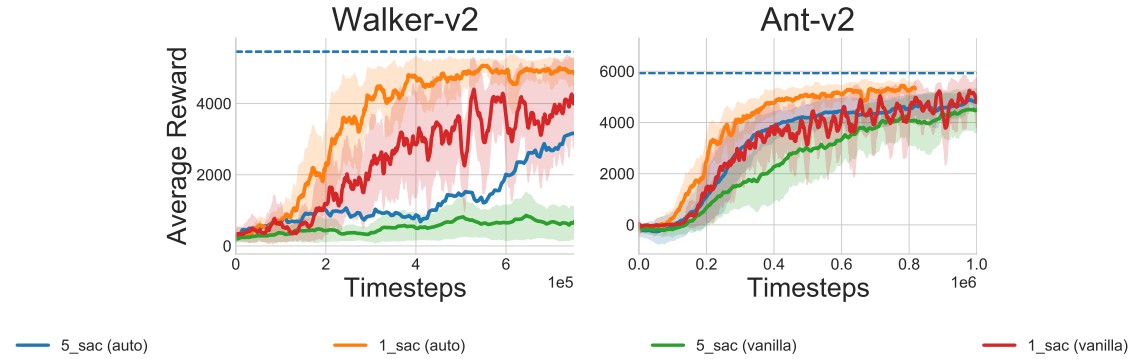

Figure 16: Small number of policy updates are useful for good learning performance in the PAL setting here.

## D.8 Sensitivity of reward range for the ranking loss $L_k$

In Section 4.2, we discussed how the scale of learned reward function can have an effect on learning performance. We validate the hypothesis here, where we set $R_{max} = k$ and test the learning performance of RANK-PAL (auto) on various different values of $k$. Our results in figure D.9 show that the hyperparameter $k$ has a large effect on learning performance and intermediate values of $k$ works well with $k = 10$ performing the best.

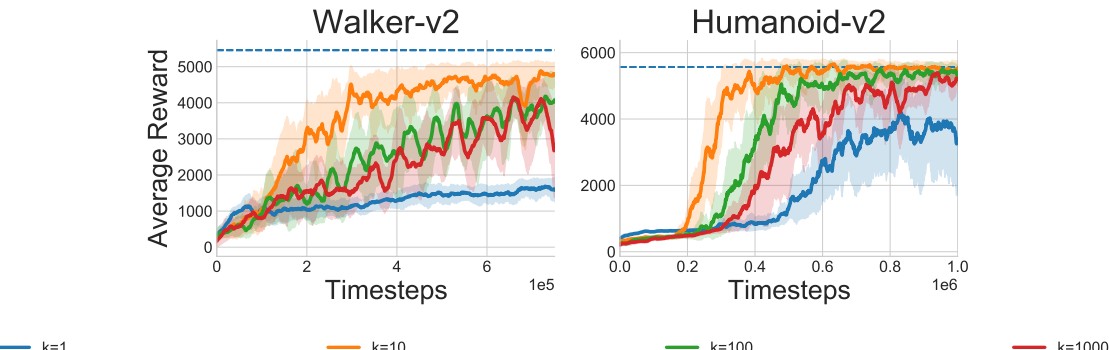

Figure 17: Intermediate values of k work best in practice.

## D.9    Effect of regularizer for rank-game

`rank-game`(auto) incorporates automatically generated rankings which can be understood as a form of regularization, particularly mixup Zhang et al. (2017) in trajectory space. In this experiment, we work in the PAL setting with ranking loss $L_k$ and compare the performances of other regularizers: Weight-decay (wd), Spectral normalization (sn), state-based mixup to (auto). Contrary to trajectory based mixup (auto) where we interpolate trajectories, in state-based mixup we sample states randomly from the behaviors which are pairwise ranked and interpolate between them.

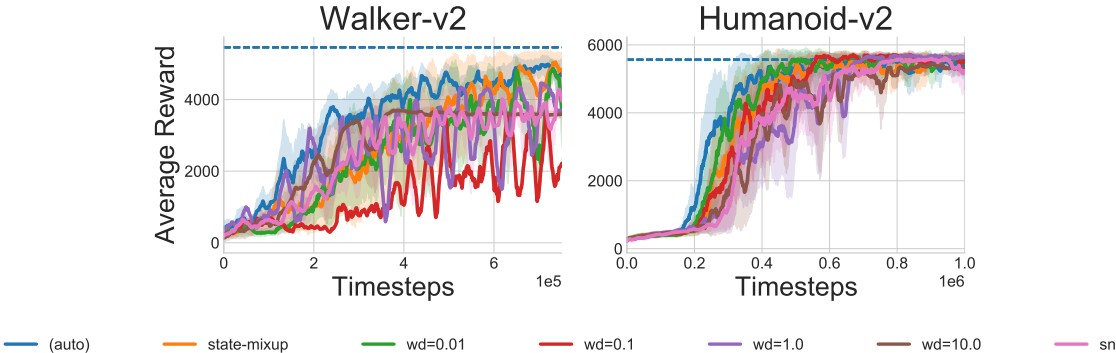

Figure 18: (auto) regularization outperforms other forms of regularization in `rank-game`

Figure 18 shows learning with (auto) regularizer is more efficient and stable compared to other regularizers.

## D.10    Ablation analysis summary

We have ablated the following components for our method: Automatically-generated rankings D.3, Ranking loss D.4, Parameterized reward shaping D.5, Stackelberg game design D.7 and range of the bounded reward D.9. Our analysis above (Figure 12,17 and 15) shows quantitatively that the key improvements over baselines are driven by using the proposed ranking loss, controlling the reward range and the reward/policy update frequency in the Stackelberg framework. Parameterized reward shaping (best hyperparameter : exp -1 compare to unshaped/linear shaping) and automatically-generated rankings contribute to relatively small improvements. We note that a *single hyperparameter* combination (Table 5) works well across all tasks demonstrating robustness of the method to environment changes.

### D.11    Varying number of expert trajectories for imitation learning

In the main text, we considered experiment settings where the agent is provided with only 1 expert trajectory. In this section, we test how our methods performs compared to baselines as we increase the number of available expert observation trajectories. We note that these experiments are in the LfO setting. Figure 20 shows that RANK-GAME compares favorably to other methods for varying number of expert demonstrations/observations trajectories.

### D.12    Robustness to noisy preferences

In this section, we investigate the effect of noisy preferences on imitation learning. We consider the setting of Section 5.2 where we attempt to solve hard exploration problems for LfO setting by leveraging trajectory snippet comparisons. In this experiment, we consider a setting similar to Brown et al. (2019) where we inject varying level of noise, i.e flip $x\%$ of trajectory snippet at random. Figure 19 shows that RANK-PAL(pref) is robust in learning near-expert behavior upto 60 percent noise in the Door environment. We hypothesize that this robustness to noise is

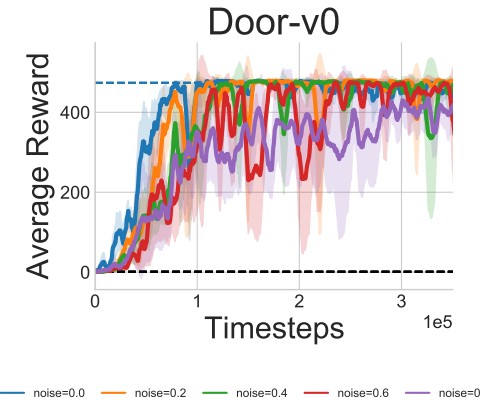

Figure 19: We investigate learning from expert observation+offline preferences where the offline preferences are noisy. RANK-PAL shows considerable robustness to noisy preferences.

possible because the preferences are only used to shape reward functions and does not change the optimality of expert.

### D.13    Learning purely from offline rankings in manipulation environments

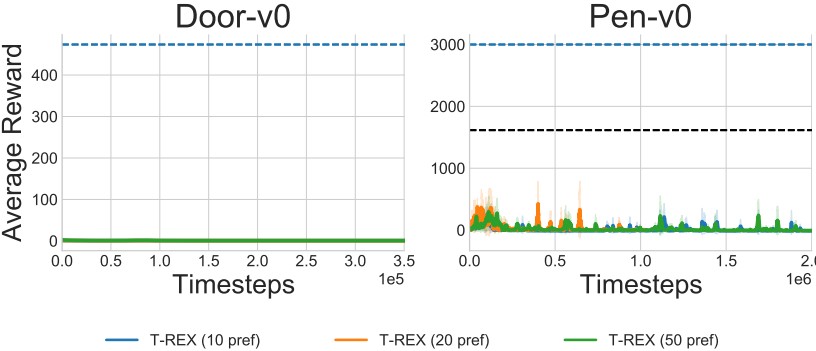

Figure 21: Testing with 10, 20 and 50 suboptimal preferences uniformly sampled from a replay buffer of SAC trained from pre-specified reward we see that TREX is not able to solve these tasks. The black dotted line shows asymptotic performance of RANK-PAL (auto) method.

In section 5.2, we saw that offline annotated preferences can help solve complex manipulation tasks via imitation. Now, we compare with the ability of a prior method—TREX (Brown et al., 2019) that learns purely from suboptimal preferences—under increasing numbers of preferences. We test on two manipulation tasks: Pen-v0 and Door-v0 given varying number of suboptimal preferences: 10, 20, 50. These preferences are uniformly sampled from a replay buffer of SAC trained until convergence under a pre-specified reward, obtained via D4RL (licensed under CC BY) .We observe in Figure 21 that T-REX is unable to solve these tasks under any selected number of suboptimal preferences.

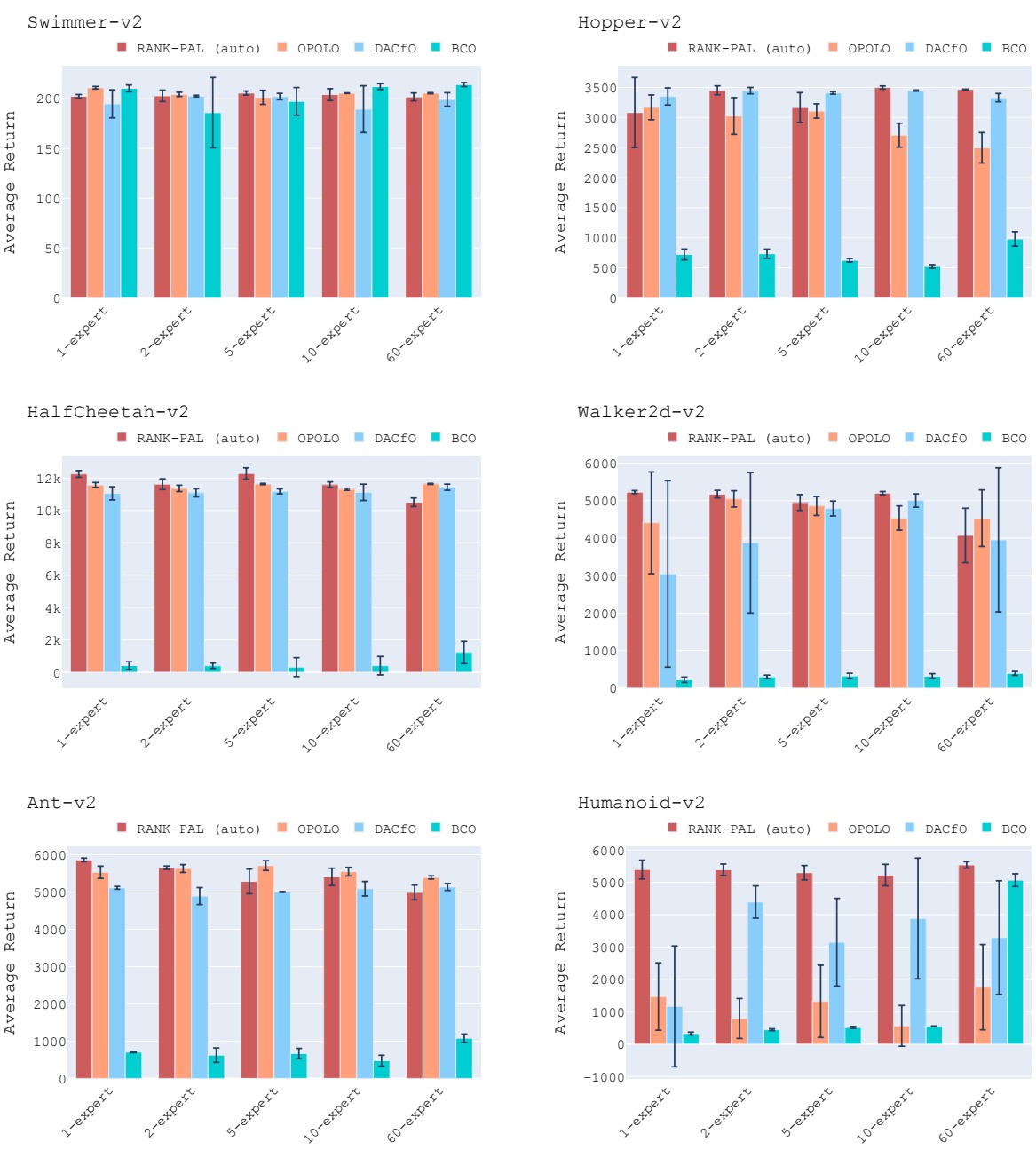

Figure 20: Performance analysis of different algorithms in the LfO setting with varying number of expert trajectories. RANK-PAL (auto) compares favorably to other methods

