# OpenReview forum: "A Ranking Game for Imitation Learning"
_TMLR — Accepted by TMLR_

### Review · Reviewer_ws4a · 2022-10-09

**Summary Of Contributions:**

This paper proposed to formulate imitation learning as a two-player ranking game (rank-game). The reward player aims to find a scalar-valued reward function that produces a separation of value $k$ between samples from any ordered pair $(\rho^{\pi^i}, \rho^{\pi^j})$ of behaviors. The policy player aims to maximize that reward. This viewpoint addresses the limitation of previous inverse reinforcement learning approaches by enabling the use of offline preferences and suboptimal behaviors in addition to expert demonstrations. The paper derives a bound on the sub-optimality of the rank-game equilibrium policy in comparison to the expert policy. Experiments in MuJoCo benchmarks show that rank-game outperforms baselines in nearly all tasks, and experiments on sparse-reward door/pen manipulation simulations show that only rank-game with the help of offline rankings can solve the tasks, while previous methods cannot.

**Audience:**

Yes

**Claims And Evidence:**

Yes

**Requested Changes:**

(Critical) Address the main apparent weakness above.

(Critical) The paper writes "The ranking loss used by the prior IRL approaches is specific to the comparison of optimal (expert) vs. suboptimal (agent) data, and precludes incorporation of comparisons among suboptimal behaviors". However, T-REX [1] explicitly allows the use of ranked suboptimal desmontrations. This paper's statements need to be revised, or it should be explained more convincingly why previous methods like T-REX have the limitations that this paper claims they do. Specifically, why can't T-REX simply assign highest preference to expert behavior.

(Critical) In Table 2, f-IRL on Walker has the highest average reward and small variance. However, in Figure 3, f-IRL is shown to have mediocre average reward and extremely large variance. This is inconsistent.

(Minor) In Section 4.2, the papers should define what it means for $\rho^{\pi} \preceq \rho^{\mu}$. Previously, the notation $\preceq$ was only define when there is a reward $f_{\pi}$ that induced the ranking $\rho^E \succeq \rho^{\pi} := E_{\rho^E}[f_{\pi}(s,a)] \geq E_{\rho^{\pi}}[f_{\pi}(s,a)]$. But in the absence of a reward $f_{\pi}$, it is not clear what $\rho^{\pi} \preceq \rho^{\mu}$ means. Perhaps the paper means the rankings are predefined by the dataset D? This can easily be clarified.

(Minor) The reference to Theorem 1 at the top of page 6 should be to Theorem 4.1?

(Minor) In section 4.2.1, the paper should clarify what a convex combination of two trajectories $\tau_i$ and $\tau_j$ means, where $\tau_i$ is a trajectory of states and actions. E.g. in discrete state-action spaces, what does a convex combination of two discrete objects mean?

(Minor) There needs to be more motivation for the choice "RAL updates the reward conservatively. This is achieved through aggregating the dataset of implicit rankings from all previous policies obtained during training". By training on data from all previous policies, RAL of course cannot adapt as fast as PAL in non-stationary environments. Why should one not simply let RAL train on the rankings generated by the current policy?


**Strengths And Weaknesses:**

Strengths

1. The proposed ranking viewpoint provides the advantage of enabling the use of suboptimal behaviors as well as expert data for IRL.
2. The experimental results show that rank-game significantly outperforms baselines on a diverse set of standard benchmarks (but see the comment below about fair comparison to T-REX).
3. The comparison between the policy-as-leader and reward-as-leader versions of rank-game on the non-stationary task are also helpful in showing the practical differences in both (but see the question below on RAL).


Weaknesses


1. The main apparent weakness, which appears to be a crucial flaw, is that the ranking loss (eq. 3) is confusing and inconsistent.
Firstly, suppose that a pair $(s,a)$ is sampled under both $\rho^{\pi^i}$ and $\rho^{\pi^j}$ when $(\rho^{\pi^i}, \rho^{\pi^j})$ appear as a ranked pair. In that case, the loss function trains the reward $R$ to map $(s,a)$ to $0$ and to $k$. That is inconsistent with the very definition of a function. Secondly, suppose we have $\pi^i \preceq \pi^j$ and $\pi^k \preceq \pi^i$. Because $\pi^i \preceq \pi^j$, samples $(s,a) \sim \pi^i$ should have reward value zero, but because $\pi^k \preceq \pi^i$, then $(s,a) \sim \pi^i$ should now have reward value k. There is no reward function that can produce both value k and value 0 for the same $(s,a) \sim \pi^i$, unless $R$ is given information about the other policy in the pair in addition to the $(s,a)$---i.e., to map some arbitrary $(s,a)$ to either $k$ or $0$ correctly, $R$ needs to know whether it came from a distribution that is ranked lower or higher in the pair $(\rho^i, \rho^j)$ that was sampled. But since that information is not provided to $R$, it is hard to see how the loss function in (eq. 3) is a consistent evaluation of $R$.
2. This paper claims that "existing IRL methods that learn from expert demonstrations provide no mechanisms to incorporate offline preferences and vice versa." However, it appears there is previous work e.g. T-REX [1], that use preferences and can accommodate the use of expert demonstrations, simply by assigning highest preference to expert demonstrations. As such, the paper's characterization of T-REX [1] in Table 2, which says T-REX is neither LfD nor LfO, appears to be incorrect. Quoting from Brown et al., T-REX is a ``reward-learning-from-observation algorithm'' that can use a set of ranked demonstrations to learn a reward function. From Section 5.2, it appears that this paper did not provide expert demonstrations to T-REX, which appears to be an unfair comparison.
3. In the second paragraph, the paper claims to address the issue of difficulty of solving an adversarial min-max game in a sample efficient way. Even though the proposed rank-game is presented as a Stackelberg game, the actual optimization procedure is still the same as in the case of a min-max game, whereby the policy and reward are optimized in an alternating and iterative manner. It is not clear, theoretically and conceptually, how the ranking game viewpoint leads to an easier optimization problem or more sample efficient learning (notwithstanding the fact that experiments show that rank-game is more sample efficient in practice).


[1] Extrapolating beyond suboptimal demonstrations via inverse reinforcement learning from observations. Brown et al. 2019.

---

> ### Author Response · Authors · 2022-10-23
> **Reply to Reviewer ws4a (1/3)**
>
> Thank you for the detailed review of our paper and the motivating comments about the approach and the empirical results. We address your concerns below. Please let us know if further clarification is needed.
>
>
> 1. **“[…]ranking loss (eq. 3) is confusing and inconsistent[…]”**.
>
> We believe there might have been a potential confusion and clarify that in our work, the reward function is a continuous map from states to scalar reward as opposed to choosing a value from the set {0,k} (see example below). In the cases when there is overlap in rankings or distribution of visited states, the reward function might not converge to either 0 or k. The ranking loss is not inconsistent as (1) The proposed ranking loss $L_k$  ***attempts*** to induce a performance gap of k between pairwise rankings of behaviors. (2) We show theoretically that this ranking loss allows us to converge to an equilibrium which solves the imitation problem. Appendix A provides more results on the obtained reward as a function of state-action visitations (Theorem A.1) as well as the cases involving multiple rankings (end of the section A).
>
> In the example proposed by the reviewer:
> We are given two preferences $\pi^k\preceq \pi^i$  and $\pi^i\preceq \pi^j$.  These preferences are given to us in the form of empirical samples of the state-action visitation distribution. Lets consider  $(s^k,a^k), (s^i,a^i),(s^j,a^j)$ be samples from respective distributions $\rho^{\pi^k},\rho^{\pi^i},\rho^{\pi^j}$ . The ranking loss function is given by : $L_k= (R(s^k,a^k)-0)^2+(R(s^i,a^i)-k)^2+(R(s^i,a^i)-0)^2+(R(s^j,a^j)-k)^2$
>
> Minimizing this loss pointwise gives us the following reward function at the three state-action pairs: $R(s^k,a^k)=0, R(s^i,a^i)=k/2, R(s^j,a^j)=k$ . Thus the preferences are respected in the learned reward function.
>
>
> 2. **“[…] T-REX, that use preferences and can accommodate the use of expert demonstrations, simply by assigning highest preference to expert demonstrations[…] the paper's characterization of T-REX [1] in Table 2, which says T-REX is neither LfD nor LfO, appears to be incorrect.[…]”**
>
> We agree with the reviewer that T-REX can accommodate the use of expert demonstrations by assigning the highest preference to expert demonstrations. We have indeed used this fact in our experiments in Section 5.2 where we ground the offline preferences with expert demonstrations. We have also updated the paper to reflect this more clearly.
>
> **On why T-REX is not a IRL method (LfO and LfD):**
>
> The problem of Inverse Reinforcement learning is defined to extract a reward function that induces the expert behavior given samples from only the optimal expert behavior— states or state-actions visitations [1]. Given access to *only* expert behavior, T-REX fails to find a reward function that induces expert behavior since it needs comparisons/preferences to be able to do reward inference. rank-game allows incorporation of “vanilla” rankings — ranking between current agent and expert in an online manner to get over this limitation. Previous works have used LfD [2,3] and LfO [4,5] to denote the imitation setting of recovering a near-expert policy from expert samples and we have followed this convention in our work. Thus, we believe the T-REX authors aptly call their method “reward-learning from observations” rather than IRL (either LfO or LfD) since (a) it cannot recover expert policy given expert demonstrations alone (b) only performs reward inference without environment interactions,  and (c) lacks guarantees of divergence matching with the expert behavior with finite samples.
>
> We also highlight that rank-game framework for imitation learning is a general-purpose method that works for both the online setting (with access to environment interactions) and offline setting. rank-game encompasses several prior imitation methods as special cases. As we discuss in Section 4.1,  T-REX is an offline reward inference method that forms a special case for rank-game when only offline rankings are available and a specific ranking-loss is used (Luce-Shepard).

---

> > ### Author Response · Authors · 2022-10-23
> > **Reply to Reviewer ws4a (2/3)**
> >
> > 3. **“[…]the actual optimization procedure is still the same as in the case of a min-max game[…]”**
> >
> > We clarify that the actual optimization is ***different*** in our method than in the case of min-max game (used in traditional adversarial imitation learning). We highlight **two key differences**: a) We use an optimization strategy of the Stackelberg formulation rather than Gradient descent ascent (GDA). GDA[12] , Best Response (BR)[11], and Stackelberg formulation[6,7,8,13] are some of the optimization techniques used to solve two-player games. Adversarial imitation learning has mostly relied on GDA whereas recent work in optimization theory and model-based RL has shown benefits of Stackelberg formulation both theoretically [6] and empirically [7, 8] over GDA and BR. b) rank-game is a general-sum game as opposed to Adversarial Imitation Learning (AIL) which are zero-sum games (min and max are over different objectives for rank-game).
> >
> > **Note on Stackelberg games:** Stackelberg formulation for general-sum games updates one player (follower) to near-convergence and then makes a conservative update for the other player (leader) under the knowledge of the follower’s strategy. We give a brief overview of Stackelberg games in Section 3. The benefits of the Stackelberg formulation over Gradient Descent-Ascent have been previously analyzed [6, 7, 8] but to the best of our knowledge, the Stackelberg formulation has not been applied to imitation learning before. Empirically, rank-game demonstrates that the optimization proposed by the Stackelberg formulation indeed leads to learning performance improvements (Appendix D.7 and D.4)
> >
> > 4. **“[…]In Table 2, f-IRL on Walker has the highest average reward and small variance. However, in Figure 3, f-IRL is shown to have mediocre average reward and extremely large variance. This is inconsistent. […]”**
> >
> > Table 2 shows the performance of the methods at 2 million timesteps as indicated in the table caption whereas Figure 3 is meant to demonstrate the sample efficiency of methods and the plots are shown up to the point where rank-game converges. In the case of Walker2d, f-IRL is sample inefficient as seen in Figure 3 (till 700000 timesteps) but converges to have the highest reward and small variance as shown in Table 2.
> >
> > 5. **[…]Perhaps the paper means the rankings are predefined by the dataset D[…]**
> >
> > Thank you for pointing this out. We have updated the paper to reflect this in Section 4.2.
> >
> > 6. **“The reference to Theorem 1 at the top of page 6 should be to Theorem 4.1?”**
> >
> > Thank you for pointing this out. We have made the notation correction in the paper.
> >
> > 7. **“[…]the paper should clarify what a convex combination of two trajectories τi and τj means[…]”**
> >
> > The automatically generated rankings are interpolations generated using convex combinations of either state  ($\lambda [s_0^{i}\,s_1^{i}..\,s_H^{i}]+ (1-\lambda) [s_0^{j}\,s_1^{j}..\,s_H^{j}]$) or state-action (LfD) trajectories ($\lambda [s_0^{i }a_0^{i }\,s_1^{i}a_1^{i }..\,s_H^{i}a_H^{i }]+ (1-\lambda) [s_0^{j}a_0^{j}\,s_1^{j}a_1^{j}..\,s_H^{j}a_H^{j}]$). In both the discrete and the continuous case, trajectories generated by the interpolation may be infeasible —  the purpose of using these rankings is to perform mixup regularization for the reward function in trajectory space. Mixup regularization[9, 10] has been shown to improve generalization and adversarial robustness. The smoothness implied by mixup regularization can better guide policy learning. We ablate the benefits from the regularization in Appendix D.9 and add clarification for trajectories in Section 4.2.1.
> >
> > 8. **“[…]Why should one not simply let RAL train on the rankings generated by the current policy?[…]”**
> >
> > In the Stackelberg framework, one player is set to be the leader and follows a conservative update strategy. For RAL, the reward is the leader and is conservatively updated by aggregating current and previous policy visitations (Algorithm 3). For RAL, if we update the reward only on the current policy, then the update is no longer conservative and the optimization reduces to GDA (similar to GAIfO or DACfO) which our experiments demonstrate are sample inefficient.

---

> > > ### Author Response · Authors · 2022-10-23
> > > **Reply to Reviewer ws4a (3/3)**
> > >
> > > References:
> > >
> > > [1]: Ng, Andrew Y., and Stuart Russell. "Algorithms for inverse reinforcement learning." *Icml*. Vol. 1. 2000.
> > >
> > > [2]: Lee, Youngwoon, et al. "Generalizable imitation learning from observation via inferring goal proximity." *Advances in Neural Information Processing Systems* 34 (2021): 16118-16130.
> > >
> > > [3]: Yang, Chao, et al. "Imitation learning from observations by minimizing inverse dynamics disagreement." *Advances in neural information processing systems* 32 (2019).
> > >
> > > [4]: Ghasemipour, Seyed Kamyar Seyed, Richard Zemel, and Shixiang Gu. "A divergence minimization perspective on imitation learning methods." *Conference on Robot Learning*. PMLR, 2020.
> > >
> > > [5]: Zhu, Zhuangdi, et al. "Off-policy imitation learning from observations." *Advances in Neural Information Processing Systems*
> > >  33 (2020): 12402-12413.
> > >
> > > [6]: Fiez, Tanner, Benjamin Chasnov, and Lillian J. Ratliff. "Convergence of learning dynamics in stackelberg games." *arXiv preprint arXiv:1906.01217* (2019).
> > >
> > > [7]: Zheng, Liyuan, et al. "Stackelberg actor-critic: Game-theoretic reinforcement learning algorithms." *Proceedings of the AAAI Conference on Artificial Intelligence*. Vol. 36. No. 8. 2022.
> > >
> > > [8]: Rajeswaran, Aravind, Igor Mordatch, and Vikash Kumar. "A game theoretic framework for model based reinforcement learning." *International conference on machine learning* . PMLR, 2020.
> > >
> > > [9]: Zhang, Hongyi, et al. "mixup: Beyond empirical risk minimization." *arXiv preprint arXiv:1710.09412*
> > >  (2017).
> > >
> > > [10]: Guo, Hongyu, Yongyi Mao, and Richong Zhang. "Mixup as locally linear out-of-manifold regularization." *Proceedings of the AAAI Conference on Artificial Intelligence*. Vol. 33. No. 01. 2019.
> > >
> > > [11]: Cesa-Bianchi, Nicolo, and Gábor Lugosi. *Prediction, learning, and games*. Cambridge university press, 2006.
> > >
> > > [12]: Li, Haochuan, et al. "On Convergence of Gradient Descent Ascent: A Tight Local Analysis." *International Conference on Machine Learning*. PMLR, 2022.
> > >
> > > [13]: Başar, Tamer, and Geert Jan Olsder. *Dynamic noncooperative game theory*. Society for Industrial and Applied Mathematics, 1998.

---

> ### Comment · Reviewer_ws4a · 2022-11-14
> **Author's rebuttal is satisfactory**
>
> The author's detailed replies have addressed the main concerns.

---

### Review · Reviewer_qjK9 · 2022-10-14

**Summary Of Contributions:**


The primary contribution of the paper is the `rank-game` framework that merges different imitation learning settings (learning from expert data and learning from preferences) by formulating it as a game between two players: a policy agent and a reward agent. The policy agent is the typical RL policy optimization agent that aims to maximize return under some reward function. Whereas, the reward agent seeks to find a reward function that satisfies the relative pairwise rankings of trajectories (or state-action distributions) as provided in the dataset.  These rankings can be the relative w.r.t. the expert trajectories (falls into ‘auto’ methodology) or offline human-labelled trajectory rankings (falls into ‘pref’ approach). The secondary contribution is a new ranking loss, $L_k$ loss, that seems to work in this new setting of using data augmentation for efficient policy optimization. The final optimization is solved using the methodology of the Stackelberg game framework.  The combination of the above two ideas leads to impressive results on a variety of synthetic mujoco domains and complex dextrous tasks.


**Audience:**

Yes

**Broader Impact Concerns:**

The authors have acknowledged the limitations of their approach and the potential side effects stemming due to a lack of guarantees for safety and robustness.

**Claims And Evidence:**

Yes

**Requested Changes:**

### Recommended changes:

- Notational inconsistencies: Sometimes the same notation is used for multiple quantities. For instance, the reward function of the MDP $R$ is also used for an unknown expert reward function (in the classical imitation learning section), and later as a ranking function in Sec. 4.2.1. I believe addressing similar notational inconsistencies will make the paper clear.

- Implications of Theorem 4.1: See my comment in the weakness section. In particular, when do the assumptions hold? How does this statement change when using other kinds of loss functions? This does not necessarily need to be studied analytically, but rather even empirical evidence on some toy domains would suffice.

###  Clarification questions:
- In Eq (2), should there be $\frac{1}{1-\gamma}$ on RHS based on definition of $J(\pi, R)$?
- Eq 4: why is $R_{max}$ in the square root term? Should it be $R^E_{max}$? Is this linked to the generalization error mentioned in Theorem 4.1's statement?
- In Fig. 3, is the cost for generating the dataset for auto also included in the figure? The algorithm needs to interact with the environment to generate the dataset so that should be included in the sample efficiency.

### Optional suggestions:
- More motivation on the setting will make the paper more accessible to readers that are not experts in the Imitation learning domain.



**Strengths And Weaknesses:**

### Strengths:

* New perspective: The idea of using both the expert data and preferences together for increasing the efficiency of learning techniques seems reasonable and useful. The proposed unified perspective of understanding various Imitation Learning algorithms from a game theoretic perspective might be interesting and inspiring to the community.

* Impressive results: The paper provides strong empirical studies, including hyper-parameter and ablation studies, that support the claims regarding the practical algorithm. The results show major improvements compared to existing methods. The authors have shown that for their setting, where both types of data modalities are present, the proposed approach is indeed promising (Sec 5.2).

* I’m not an expert in Imitation Learning but it seems that the authors have conducted a proper discussion of the related work.

### Weaknesses:

The biggest weakness of the work is the lack of clarity in Section 4.2.  While the motivation of the proposed loss is that it facilitates better incorporation of rankings over suboptimal trajectories, can you highlight why this is a principled loss?  I appreciate the theoretical result that tells the performance with the expert is bounded, but it seems like the upper bound is quite loose. The first term in the result is the maximum attainable return under the expert reward function $\frac{R^E_{max}}{1-\gamma}$, whereas the multiplicative term mostly depends on hyper-parameter $k$ as the other quantities that are not defined before Theorem 4.1. What is the generalization error and when will it be bounded by the given quantity in Theorem 4.1? It looks like the user should set $k \rightarrow \infty$ to reduce the gap, however, from the experiments (Fig. 17) it seems like intermediate $k$ works better than large $k$. Moreover, does the equilibrium always exist? Unfortunately, the theoretical result in the present form makes things rather confusing.

I understand that lot of these details might already be present in Appendix A, however, Sec 4.2 should be able to convey these results in a standalone manner without expecting the reader to go through Appendix A and D.

---

> ### Author Response · Authors · 2022-10-23
> **Reply to Reviewer qjK9 (1/2)**
>
> We thank the reviewer for the detailed review of our paper. We are motivated that you found the new perspective interesting and inspiring, and the empirical results impressive.  We address your concerns below. Please let us know if further clarification is needed.
>
> 1. **“[…]can you highlight why this is a principled loss? […] but it seems like the upper bound is quite loose. […]What is the generalization error and when will it be bounded by the given quantity in Theorem 4.1? It looks like the user should set k→∞ to reduce the gap, however, from the experiments (Fig. 17) it seems like intermediate k works better than large k. Moreover, does the equilibrium always exist?**
>
> a. **“[…]can you highlight why this is a principled loss? […]"**
>
> The ranking loss is principled as (1) at equilibrium, it induces a policy that is close to the expert’s policy and (2) it learns a reward function that attempts to map returns to behaviors according to their preferences (Appendix D.6). Theorem 4.1 demonstrates that this is indeed the case and at equilibrium, the performance of the policy obtained is bounded with the performance of the expert policy.
>
> b. **“[…] but it seems like the upper bound is quite loose. […]”**
>
> We highlight that the bound is not loose as the tightest known bounds in imitation learning come from the IRL setting (also referred as reward moment matching) and are also lower bounded by $\frac{R_{max}}{1-\gamma}.$  [1] presents an MDP construction to demonstrate this tightness in Lemma 2, and similar bounds can also be found in [2]. The realizability assumption ensures that expert reward function lies in the agent reward function class, i.e $R^E_{max}\le R_{max}$.  We also note that a number of previous works consider $R\in [0,1] $ which hides the $R_{max}$ term from the upper bound.
>
> c. **“What is the generalization error and when will it be bounded by the given quantity in Theorem 4.1? It looks like the user should set k→∞ to reduce the gap, however, from the experiments (Fig. 17) it seems like intermediate k works better than large k.”**
>
> The ranking loss generalization error will be bounded by $\epsilon_r$ with high probability. Unlike previous works [1,2], we consider the finite samples at each reward learning iteration for our analysis of the equilibrium point. The ranking loss with finite samples incurs a generalization error that can be characterized by concentration bounds such as Hoeffding’s.
>
> Why shouldn’t we set $k$ to the $\infty$?
>
> Our reward function is assumed to bounded by $[0,R_{max}]$ and the performance gap is set to be k. Note that our analysis assumes $k\in[0,R_{max}]$ as well since the ranking loss cannot assign a performance gap that is greater than range of the reward function class. We have updated section 4.2 to clarify this. Hence, $k\to\infty$ is not possible in this setting.
>
> Why shouldn’t we set $k$ to the largest value $R_{max}$? We include this in our discussion in Section 4.2, paragraph 3. Prior work[3] has found that a larger $k$, consequently a larger reward scale, can make the policy optimization landscape less smooth and decrease learning efficiency. We observe this in our empirical studies (in Appendix D.9) which shows that an intermediate $k$ often works well in practice.
>
> d. **“Moreover, does the equilibrium always exist? ”**
>
> We answer this question affirmatively in Section 4.2 and Appendix A by showing that any given iteration, the policy optimization in rank-game is equivalent to minimizing a particular $f$-divergence with the expert behavior and the $f$-divergence is minimized *only* when the two distributions match. Although this equilibrium exists, whether we can theoretically reach the equilibrium with non-convex optimization methods like Adam, is a different question that is beyond the scope of this work. We  present a regret analysis in Lemma A.2, which shows how an optimistic policy optimization procedure can achieve sublinear regret in this setting.
>
>
> We point the reviewer to the updated Section 4.2 where we elaborate on the terms in Theorem 4.1 to provide better intuition.

---

> > ### Author Response · Authors · 2022-10-23
> > **Reply to Reviewer qjK9 (2/2)**
> >
> > 2. **“[…]In particular, when do the assumptions hold? How does this statement change when using other kinds of loss functions?[…]”**
> >
> > As we point out above our assumptions are quite weak — the upper bound depends on controllable hyperparameters ($k,R_{max}$) along with finite sample errors that are characterized by concentration bounds. We also take into account the errors in the policy learning procedure for Deep RL that may arise due to the value approximation error, finite samples, and various biases[4,5,6] to bring our analysis closer to the real setting. In Appendix D.4, we discuss the empirical differences in learning performance when the ranking loss is replaced by the supremum loss (a variational form of the $f$-divergence loss function used in GAIfO, DACfO).
> >
> > 3. **Notational inconsistency**
> >
> > We thank the reviewer for pointing it out. We have updated the paper to reflect the expert’s unknown reward function $R_{gt}$ for classical imitation learning in Section 3.
> >
> > 4. **[…]In Eq (2), should there be[…]**
> >
> > Thank you for pointing this out. We have corrected the missing term in the equation.
> >
> > 5. **“[…]Rmax in the square root term? Should it be[…] Is this linked to the generalization error mentioned in Theorem 4.1's statement?[…]"**
> >
> > We elaborate on the reason behind the presence of $R^E_{max}$ as an outer scaling factor in point 1 above. Yes, the reviewer correctly states that $R_{max}$ in the square root term is due to scaling of the generalization error. We assume a canonical generalization error $\epsilon_r$ for reward function class in range [0,1] and scale it according to the maximum value the learned reward function can take.
> >
> > 6. **“[…]In Fig. 3, is the cost for generating the dataset for auto also included in the figure?[…]”**
> >
> > Yes. Generating the intermediate rankings in the auto-dataset is offline and does not require environment interactions. The cost for all environment interactions during the learning process is included in Figure 3.
> >
> > References:
> > [1] Swamy, Gokul, et al. "Of moments and matching: A game-theoretic framework for closing the imitation gap." *International Conference on Machine Learning*. PMLR, 2021.
> >
> > [2]: Xu, Tian, Ziniu Li, and Yang Yu. "Error bounds of imitating policies and environments." *Advances in Neural Information Processing Systems* 33 (2020): 15737-15749.
> >
> > [3]: Henderson, Peter, et al. "Deep reinforcement learning that matters." *Proceedings of the AAAI conference on artificial intelligence*. Vol. 32. No. 1. 2018.
> >
> > [4]: Fujimoto, Scott, Herke Hoof, and David Meger. "Addressing function approximation error in actor-critic methods." *International conference on machine learning*. PMLR, 2018.
> >
> > [5]: Thrun, Sebastian, and Anton Schwartz. "Issues in using function approximation for reinforcement learning." *Proceedings of the 1993 Connectionist Models Summer School Hillsdale, NJ. Lawrence Erlbaum*. Vol. 6. 1993.
> >
> > [6]: Fu, Justin, et al. "Diagnosing bottlenecks in deep q-learning algorithms." *International Conference on Machine Learning*. PMLR, 2019.

---

### Review · Reviewer_UMKz · 2022-10-17

**Summary Of Contributions:**

The paper introduces a ranking framework for imitation learning. The frameworks consists of an agent that attempts to maximize rewards estimation via another ranking agent.

**Audience:**

Yes

**Broader Impact Concerns:**

I have no broader impact concerns.

**Claims And Evidence:**

Yes

**Requested Changes:**

* Update algorithm 1 to clarify the policy update step.
* Update the paper with infomation about baselines and datasets.

**Strengths And Weaknesses:**

# Strengths
* The paper demonstrates good empirical results on several datasets.
* Ranking and incorporating references is a topic interesting to the deep reinforcement learning community.

# Weaknesses
* The paper has limited novelty. The method can be seen as an instantiation of GAILfO combined with the LSGAN loss function.
* The writing needs to be improved. The paper is hard to follow. In particular, it is unclear whether the authors consider LfD as an imitation learning of a reinforcement learning setup. Several previous papers on learning from demonstrations (for example, DQfD and DDPGfD) considered the setting where the rewards are available.
* From algorithm 1. It's not clear how step 5 is performed. Do you use directly SAC to learn a policy based on these rewards? Do you perform any modifications for SAC?
* The paper doesn't describes datasets used for evaluation and information about the baseline implementation is missing.

---

> ### Author Response · Authors · 2022-10-23
> **Reply to Reviewer UMKz**
>
> We thank the reviewer for their review of our paper and the motivating comments about the relevance of the approach and the empirical results. We address your concerns below. Please let us know if further clarification is needed.
>
> 1. **‘The paper has limited novelty. The method can be seen as an instantiation of GAILfO combined with the LSGAN loss function’**
>
> We thank the reviewer for pointing out LSGAN which we now explicitly acknowledge in the paper (Appendix B.1). However while the proposed ranking loss is similar in spirit,  the following components remain novel:
>
> a. The analysis of the ranking loss in the sequential decision-making setting for imitation learning. LSGAN deals with the supervised learning setting.
> b. The proposed ability of the ranking loss to incorporate suboptimal preferences to boost imitation learning performance. LSGAN does not consider using suboptimal data or comparisons within them.
> c. We obtain significant gains from using the Stackelberg formulation for optimizing the ranking loss in the imitation setting. LSGAN does not investigate any such optimization methods for solving their min-max game.
> d. The empirical evaluation of the ranking loss's utility for the imitation learning setting, with and without rankings between suboptimal behaviors, across manipulation and locomotion domains.
>
> 2. **‘[…]it is unclear whether the authors consider LfD as an imitation learning of a reinforcement learning setup.[…]’**
>
> We add a discussion in the introduction section of the paper to clarify the setting. In the paper, we have strived to follow the established convention from previous papers to refer to different settings in reinforcement learning and imitation learning. Prior work [1,2,3,4] has used LfD and LfO to refer to imitation learning settings, and RLfD [5,6,7] to refer to the reinforcement learning setting where the expert reward function is known.
>
> 3. **“From algorithm 1. It's not clear how step 5 is performed.”**
>
> We use Soft actor-critic without any modification for step 5. We note that the reward obtained by SAC in this setting comes from the learned reward function. We state this in the first paragraph of Section 5 of the paper as well as in Appendix C (Algorithms 2 and 3).
>
> 4. **“The paper doesn't describes datasets used for evaluation and information about the baseline implementation is missing.”**
>
> The details about the environments/datasets (MuJoCo, D4RL) and baselines (OPOLO, f-IRL, IQLearn, DACfO, GAIfO, BCO) can be found in Appendix C and are pointed to in Section 5.
>
> References:
>
> [1]: Ghasemipour, Seyed Kamyar Seyed, Richard Zemel, and Shixiang Gu. "A divergence minimization perspective on imitation learning methods." *Conference on Robot Learning*. PMLR, 2020.
>
> [2] Lee, Youngwoon, et al. "Generalizable imitation learning from observation via inferring goal proximity." *Advances in Neural Information Processing Systems* 34 (2021): 16118-16130.
>
> [3]: Yang, Chao, et al. "Imitation learning from observations by minimizing inverse dynamics disagreement." *Advances in neural information processing systems* 32 (2019).
>
> [4]: Zhu, Zhuangdi, et al. "Off-policy imitation learning from observations." *Advances in Neural Information Processing Systems* 33 (2020): 12402-12413.
>
> [5]: Jing, Mingxuan, et al. "Reinforcement learning from imperfect demonstrations under soft expert guidance." *Proceedings of the AAAI conference on artificial intelligence*. Vol. 34. No. 04. 2020.
>
> [6]: Brys, Tim, et al. "Reinforcement learning from demonstration through shaping." *Twenty-fourth international joint conference on artificial intelligence*. 2015.
>
> [7]: Zhang, Haoran, et al. "Self-Guided Actor-Critic: Reinforcement Learning from Adaptive Expert Demonstrations." *2020 59th IEEE Conference on Decision and Control (CDC)*. IEEE, 2020.

---

### Author Response · Authors · 2022-10-23
**General Response**

We thank all reviewers for their constructive feedback. We are motivated that the reviewers found the unified perspective interesting and inspiring [**ws4a, qjK9**], and the empirical results thorough and impressive [**ws4a, qjK9, UMKz**]. We have addressed each review individually and will use this comment to summarize the most important changes and additions. The updates to the paper are highlighted in red.

1. Increased clarity and intuition for Theorem 4.1 in Section 4.2
2. Added clarification to distinguish LfD from RLfD setting in Introduction.
3. Corrected notational issues pointed out by reviewers ****ws4a, qjK9****

---

### Decision · Action_Editors · 2023-01-08

**Recommendation:** Accept as is

**Comment:**

Reviews are largely in concensus for acceptance, as the authors also had satisfying rebuttals.

Reviewer UMKz expressed "The paper has limited novelty. The method can be seen as an instantiation of GAILfO combined with the LSGAN loss function". While ranking loss is a simple common objective, LSGAN does not consitute a major conflict with this work as it deals with supervised learning. Therefore, this does not invalidate the novelty of the work. Focused discussion and experimentation of IL+ranking in sequential decision making setting still makes the work valuable for the audience.

Minor comments:
- improve writing clarity: Table 1, Section 4.1, Eq 1 etc: write down exact mathematical formula for each ranking loss in one place.
- additional experiments: auto-generated preference orders through mixup likely do not scale to high-dimensional state / action (image / texts). Additional experiments to show that the method scales would be helpful.

**Audience:**

Yes. The paper connects literatures in RL from preferences with imitation learning well with thorough discussions on how they can be unified mathematically, and how they decompose and relate to prior work (Table 1). It's a well writen educational paper, with proofs of concepts for empirical applications. Some algorithmic deep RL researchers would find values from this work.

**Claims And Evidence:**

Yes.

Claims are:
1) propose a new method for imitation learning based on ranking game
- this is well supported through derivations and theoretical analyses. Table 1 also helps to clarify its differentiations from prior work.

2) "achieves state-of-the-art sample efficiency"
- well supported, e.g. Figure 3. Though please understand the general skepticism around reproducible improvements on these simple environments.

3) "can solve previously unsolvable tasks in the Learning from Observation (LfO) setting"
- this is also supported and probably the most convincing empirical result of the paper, e.g. Figure 4. Though it would be important to qualify that to achieve Figure 4, ranking labels based on ground-truth reward is crucial, as opposed to "auto".

---

> ### Author Response · Authors · 2023-01-16
> **Thank you!**
>
> Thank you for the comments! We have updated Section 4.1 with an exact mathematical formula for each ranking loss in one place. We believe that extending this method to large observation spaces (e.g. images) would require combining the auto-generated preferences procedure with a suitable representation learning method that would allow us to generate preferences in that representation space, and hence we leave this exploration to future work.
>
> We appreciate the effort reviewers and the action editor made on improving the clarity and quality of the paper! Thank you again for a great review process.